# Temperature and insulin signaling regulate body size in *Hydra* by the Wnt and TGF-beta pathways

Benedikt M. Mortzfeld [1,3,5], Jan Taubenheim [1,4,5], Alexander V. Klimovich [1], Sebastian Fraune[1,4], Philip Rosenstiel[2] & Thomas C.G. Bosch[1]

How multicellular organisms assess and control their size is a fundamental question in biology, yet the molecular and genetic mechanisms that control organ or organism size remain largely unsolved. The freshwater polyp *Hydra* demonstrates a high capacity to adapt its body size to different temperatures. Here we identify the molecular mechanisms controlling this phenotypic plasticity and show that temperature-induced cell number changes are controlled by Wnt- and TGF-β signaling. Further we show that insulin-like peptide receptor (INSR) and forkhead box protein O (FoxO) are important genetic drivers of size determination controlling the same developmental regulators. Thus, environmental and genetic factors directly affect developmental mechanisms in which cell number is the strongest determinant of body size. These findings identify the basic mechanisms as to how size is regulated on an organismic level and how phenotypic plasticity is integrated into conserved developmental pathways in an evolutionary informative model organism.

[1] Zoological Institute, Christian-Albrechts University Kiel, Am Botanischen Garten 1-9, 24118 Kiel, Germany. [2] Institute of Clinical Molecular Biology, Christian-Albrechts University Kiel, University Hospital Schleswig-Holstein, Rosalind-Franklin-Straße 12, 24105 Kiel, Germany. [3] Present address: Department of Bioengineering, University of Massachusetts Dartmouth, 285 Old Westport Rd, Dartmouth, MA 02747, USA. [4] Present address: Institute for Zoology and Organismic Interactions, Heinrich-Heine University Düsseldorf, Universitätsstraße 1, 40225 Düsseldorf, Germany. [5] These authors contributed equally: Benedikt M. Mortzfeld, Jan Taubenheim. Correspondence and requests for materials should be addressed to T.C.G.B. (email: tbosch@zoologie.uni-kiel.de)

Body size is assumed to be a species-specific feature with physical/physiological restrictions and major implications for the fitness of an organism[1]. Yet, tremendous variations in size are reported for individuals of the same species[2]. Assuming a common genetic background for a species poses the question, which factors determine the size of an individual organism[3].

On a cellular level, only three basic factors contribute to size: the growth period (developmental time), the amount of cells gained during this period (growth rate) and the individual cell size[4]. Genetic factors are considered to be a major determinant of these factors widely conserved genes and pathways including homeobox genes[5], components of the Insulin[6], Wnt[7], or TGF-β[8] signaling pathways, contribute to size determination.

Only 27% of human height variation can be explained by heritability, leaving over 70% of height control to environmental factors and phenotypic plasticity[9]. Environmental factors like nutrition[10,11] or the surrounding temperature are the best studied which affect size determination in almost all organisms[12,13]. Since such phenotypic plasticity can increase organism survival under varying conditions, it is hypothesized to contribute to evolutionary processes and speciation[14,15].

However, even though alterations of environmental factors or conserved pathways seem to affect body size, it is not well understood, (i) how environmental cues affect genetic programs, (ii) how size is measured by cells or tissue during growth, and (iii) by what processes termination of growth is initiated. Indeed, understanding of these processes promises not only to contribute to the comprehension of ecological and developmental interactions on a molecular level, but may have major implication for evolutionary concepts.

In most organisms, developmental programs determine an irreversible size phenotype[15], while cnidarians such as Hydra are extremely flexible throughout their life history. Adult Hydra polyps possess many embryonic characteristics such as constant cell proliferation and continuous developmental patterning, while being non-senescent and clonal in growth[16]. The morphological appearance is determined by the two unipotent epithelial stem cell lines, which have a remarkably robust cell cycle[17] and whose proliferation is unaffected even by harsh environmental conditions like starvation[18]. Together with the continuously active, Wnt-mediated patterning process along the single body axis these embryonic features are the basis of Hydra's regeneration capacity[19].

Applying different environmental cues to clonal animals which are both members of an evolutionary ancient taxon and still in a developmental life stage provides the rare opportunity to study phenotypic plasticity of size and intrinsic size determination within a single genotype and thus allows to reveal conserved molecular mechanisms underlying individual size control by environmental and genetic triggers. In this study we show, that insulin and FoxO signaling together with temperature cues are integrated into the same signaling machinery of the conserved Wnt and TGF-β pathways. Wnt is directly affecting the expression of TGF-β pathway components, which controls the timing of asexual reproduction and forms a developmental switch between the growth and reproductive phase of the animals. This developmental switch eventually determines the cell number and thus the size of Hydra.

## Results

**Temperature-induced phenotypic plasticity of body size.** Hydra can be cultured at a wide temperature range[20]. Clonal polyps reared at different temperatures grow to a temperature-dependent size (Fig. 1a): At low temperatures (8 °C, 12 °C) animals are visibly larger than at higher temperatures (18 °C, 22 °C). To quantify these size differences, we determined the total epithelial cell number per polyp with a small bud protrusion, further referred to as maximum size, by flow cytometry. Epithelial cells of dissociated polyps were gated using forward and side scatter (Supplementary Fig. 1, Supplementary Table 1) revealing that Hydra's epithelial cell number is temperature dependent, with the lowest number of epithelial cells ($12,963 \pm 3555$; mean ± SD; $n = 11$) at 22 °C. With decreasing temperature the number of epithelial cells increases by 83% (12 °C; $23,698 \pm 3537$; $n = 18$) with no further gain in cell number at 8 °C ($21,441 \pm 8565$; $n = 16$) (Fig. 1b). Further, we found a linear increase of cell size with decreasing temperatures, ranging from $25.0 \pm 0.6$ μm (22 °C; $n = 11$) to $30.7 \pm 0.4$ μm (8 °C; $n = 16$) (Fig. 1c). To test the dependence of the size on the cell proliferation rate we measured BrdU labeling rates at 12 °C and 18 °C rearing temperature, but observed no difference in labeling efficiency after a 3 h BrdU pulse (Supplementary Fig. 2a), suggesting that the cell cycling rate was not affected. Remarkably, 72 h labeling with BrdU resulted in decreased BrdU incorporation at 12 °C, suggesting an elongated cell cycle in larger animals. Since time between cell divisions usually correlates with the cell size, the slowed cell cycle at lower temperatures seems to account for larger epithelial cells[21]. However, the reduced proliferation rate cannot account for higher epithelial cell numbers and other mechanisms should be involved.

In order to resolve this discrepancy and to elucidate the molecular mechanisms which underlie the size determination in Hydra, we next compared transcriptomes of polyps reared at different temperatures. Using 25 RNA sequencing libraries we assembled a new transcrpitome[22]. BUSCO[23] analysis revealed a transcriptome completeness of 86.4% and mapping rates for all 60 samples reached 95–98%. Further statistics are summarized in Supplementary Table 2. After annotation we concluded that 58% of the open reading frames (ORFs) contain conserved motifs, while 42% show no homologous regions to various databases (including SMART, Pfam, PANTHER, KEGG) and therefore can be considered as orphan genes (Supplementary Fig. 3)[24].

Next we performed a differential expression (DE) analysis comparing the transcriptomes of clonal animals at different rearing temperatures in a pairwise manner (8 °C vs. 12 °C, 12 °C vs. 18 °C, and 18 °C vs. 22 °C). Since the temperature shift from 8 °C to 12 °C did not decrease the cell number per polyp significantly (Fig. 1b, c), we only included differentially expressed genes from the comparisons associated with a cell number difference (12 °C vs. 18 °C and 18 °C vs. 22 °C, Fig. 1d, bold, total number of 1580). While the expression of most gene clusters increased or decreased in a gradual manner (Fig. 1e, framed green), the expression of some genes responded solely to one condition (Fig. 1e, framed yellow). Moreover, especially genes specific for the 12 °C vs. 18 °C comparison acted biphasically as a switch between lower and higher temperatures (Fig. 1e, framed red). 205 DE genes were shared between the temperature switches 12 °C vs. 18 °C and 18 °C vs. 22 °C making them promising candidates to control cell number and thereby maximum body size.

Within the DE gene clusters we identified several conserved genes that could be involved in size regulation, according to their role in model organisms, by playing a role in apoptosis (Casp3)[25], proliferation (fibroblastic growth factor receptor, R-Ras)[26] and developmental processes (Notum, Notch, Thrombospondin 1, Calmodulin)[27–30] (Supplementary Data 1). The DE genes from the different temperature comparisons and putative candidates for size determination showed neither an enrichment for a particular pathway nor an obvious functional overlap. However, temperature is not only known to affect size but also metabolic pathways and physiology[31]. Consequently, we suspected size

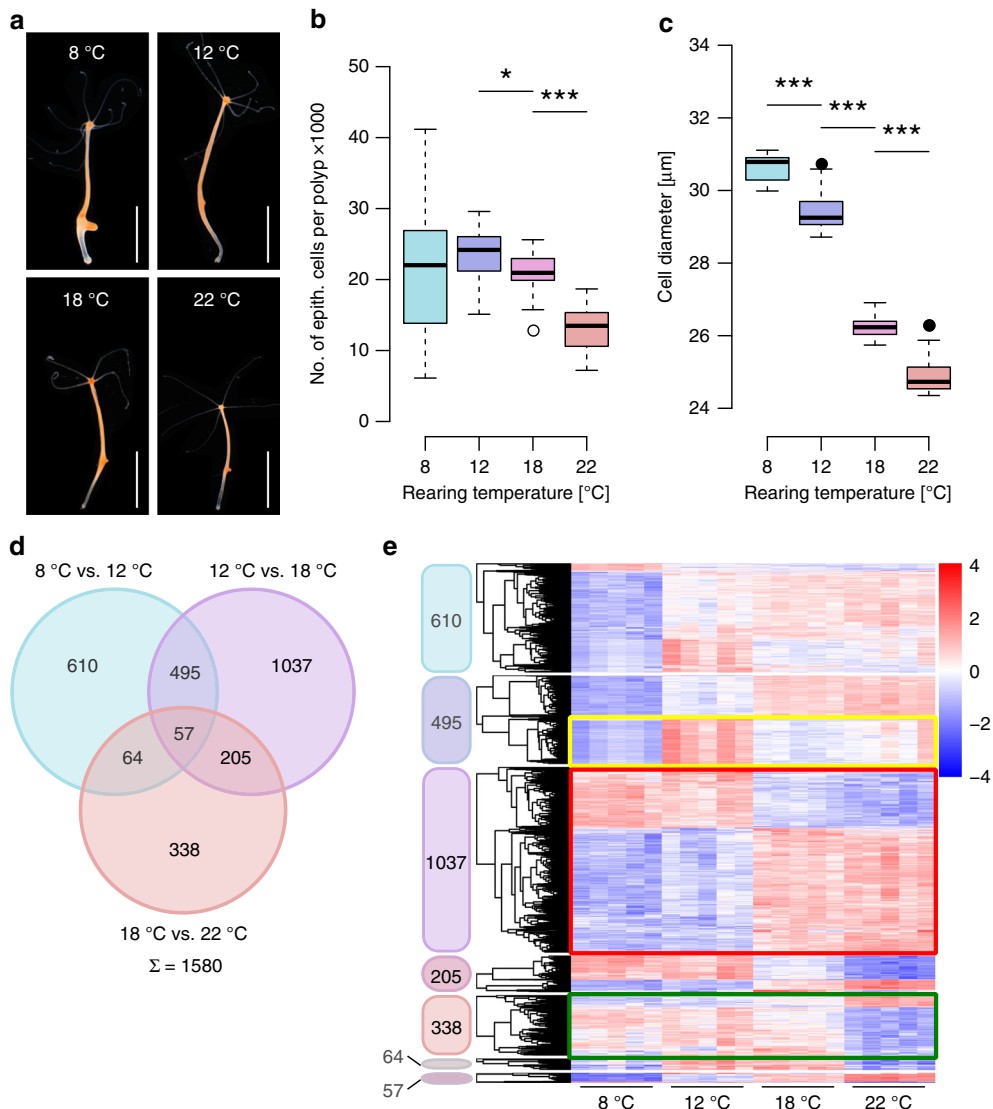

**Fig. 1** Temperature-induced phenotypic plasticity. **a** *Hydra* polyps reared at 8 °C, 12 °C, 18 °C and 22 °C at maximum polyp size with first bud protrusion. Scale bar: 5 mm. **b**, **c** Clonal *Hydra* polyps reared at temperatures from 8 °C to 22 °C showed decreasing cell numbers and cell size of epithelial cells at maximum size. Boxplots show median (horizontal line), lower and upper quantile (box), lower and upper 1.5 times interquartile range (whiskers) and outliers (points). $n = 16$ samples/1043640 cells (8 °C), 18 samples/1376686 cells (12 °C), 18 samples/1318035 cells (18 °C), 11 samples/479968 cells (22 °C), $*p \leq 0.05$, $***p \leq 0.001$ (U-test + FDR-correction). **d** Venn diagram showing number of differentially expressed genes comparing the different temperature treatments (8 °C vs. 12 °C, 12 °C vs. 18 °C, 18 °C vs. 22 °C). A total of 1580 genes (bold) are associated with a significant change in maximum size by temperature shift. **e** Heatmaps representing the relative expression level of the differentially expressed genes (rows) from the venn diagram over the different temperatures. Genes were clustered hierarchically per heatmap. Expression values were $\log_2$-transformed and median-centered by transcript. $n = 5$ libraries (Wald-Test + FDR-correction)

determination to be a holistic feature of organisms and a general, comprehensive mechanism which comprises both, phenotypic plasticity, as well as the intrinsic genetic size component.

**INSR-knockdown induces larger polyp size.** Insulin signaling is known to be involved in size regulation and cognition of environmental signals in diverse organisms[6]. We therefore hypothesized that the insulin receptor (INSR) and its transcription factor forkhead box protein O (FoxO) are promising candidates for being involved in driving environmentally dependent size regulation in *Hydra*. For functional analysis of their involvement in environmental signaling, development and size determination, we generated transgenic animals carrying an shRNA construct against the putative *insR* gene[32] to disrupt INSR-dependent

signaling (Supplementary Fig. 4a,b). Microinjection of the hairpin DNA into early embryos resulted in mosaic transgenic hatchlings[33] and selection for transgenic cells in the ectodermal and endodermal cell line during the asexual reproduction of *Hydra* eventually resulted in completely transgenic animals. At the same time, we selected for non-transgenic cells to obtain animals with the same genotype, but without expressing the hairpin construct, which served as a control line for the INSR-knockdown (INSR-KD) animals in all further experiments[34]. The INSR-KD resulted in a significant increase of the maximum polyp size (Fig. 2a). Consistently, the epithelial cell number per polyp at 18 °C (9903 ± 2447; $n = 20$; to 11,498 ± 2950; $n = 20$) and 22 °C (7328 ± 1154; $n = 16$; to 8876 ± 1237; $n = 16$) increased significantly by 16% and 21%, respectively, compared to its control line (Fig. 2b, Supplementary Fig. 5a, b and Supplementary

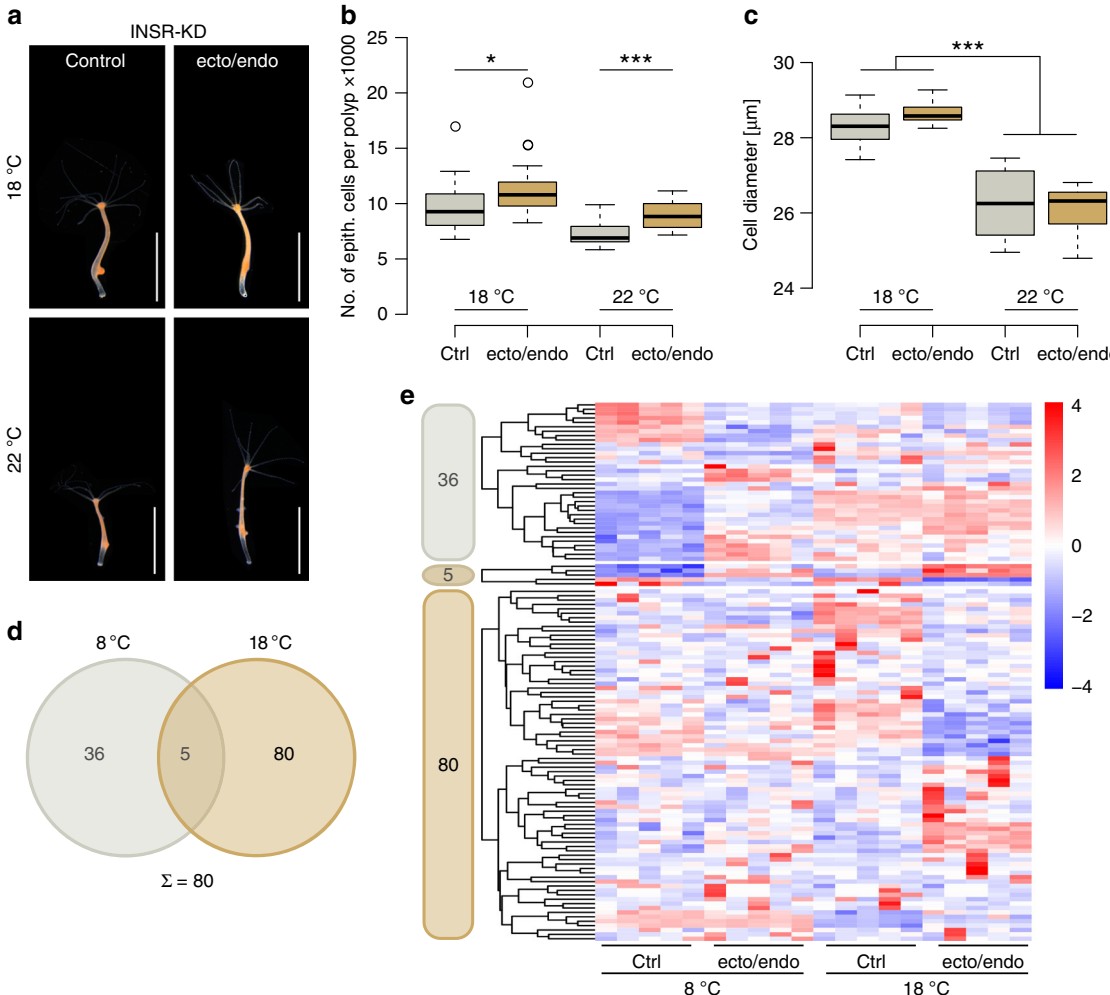

**Fig. 2** INSR-KD induces larger polyp size. **a** Pictures of control and INSR-KD polyps at 18 °C and 22 °C with bud protrusion. Scale bar: 5 mm. **b** Control and INSR-KD polyps reared at temperatures from 18 °C to 22 °C. The INSR-KD results in larger maximum size with more epithelial cells per polyp. Note, that KD animals still increase in size at lower temperatures. **c** Cell size is not affected by the KD of the INSR, though cell size is still temperature sensitive. Boxplots show median (horizontal line), lower and upper quantile (box), lower and upper 1.5 times interquartile range (whiskers), and outlier points. $n$ = 20 samples/368394 cells (18 °C ctrl), 20 samples/422092 cells (18 °C ecto/endo), 16 samples/226238 cells (22 °C ctrl), 16 samples/272842 cells (22 °C ecto/endo), *$p \leq 0.05$, ***$p \leq 0.001$ ($U$-test + FDR-correction). **d** Venn diagram showing the overlap of differentially expressed genes by INSR-KD at 8 °C and 18 °C. 80 genes (bold) are associated with increased size at 18 °C (see Supplementary Fig. 5a, b). **e** Heatmaps representing the relative expression level of the differentially expressed genes (rows) from the venn diagram of the INSR-KD animals. Genes were clustered hierarchically per heatmap. Expression values were log$_2$-transformed and median-centered by transcript. $n$ = 5 libraries (Wald-test + FDR-correction)

Fig. 6a). We found no increase in cell size caused by the INSR-KD, neither at 18 °C nor at 22 °C (Fig. 2c). Since insulin signaling is known to control cell proliferation[6], we next compared BrdU labeling rates in INSR-KD and control animals and found no differences in cell proliferation between transgenic and control polyps (Supplementary Fig. 2b). Therefore, differences in cell cycle could again not account for increased number of epithelial cells per polyp. Since growth is not only determined by the cell proliferation, but also by the length of the growth period, we measured the developmental time of a freshly detached bud from the mother polyp, until it reached its adult size demarcated by the induction of a bud protrusion ($\Delta T_1$). INSR-KD animals showed a significantly prolonged growth period explaining the increase in epithelial cell number per polyp (Supplementary Fig 6b, c). Remarkably, INSR-KD animals were still responsive to temperature changes and a similar gradient of size as in wild type controls with significant differences in cell number between 12 °C to 18 °C and 18 °C to 22 °C could be observed (Supplementary Fig. 5a, b). Therefore, we concluded that the reproducible

phenotypic plasticity observed when the animals were exposed to different temperatures is not mediated by INSR-dependent signaling. Furthermore, we concluded that the change in cell size is a temperature dependent feature and focused on changes in cell number to elucidate its genetic and environmental regulation.

Next we compared the transcriptomes of control and epithelial INSR-KD animals reared at 18 °C and 8 °C to identify genes that account for the size difference independently of temperature but dependent on INSR signaling. Comparing INSR-KD and control polyps at 18 °C and 8 °C, we utilized the animals reared at 8 °C as control, since no significant size difference was observed at lower temperatures (Supplementary Fig. 5a, b). We found 41 DE genes at 8 °C rearing temperature (5 of these overlapping with DE genes at 18 °C), which included genes in metabolic processes (PPP1R3, ETFDH, TTC19, and MPAO) and for membrane stabilization (PKD1L2 and ATP1A) suggesting distinct effector genes of the insulin pathway at 8 °C and 18 °C (Supplementary Data 1). The larger fraction (80, bold) of exclusively DE genes was found when comparing the transcripts of INSR-KD polyps and control

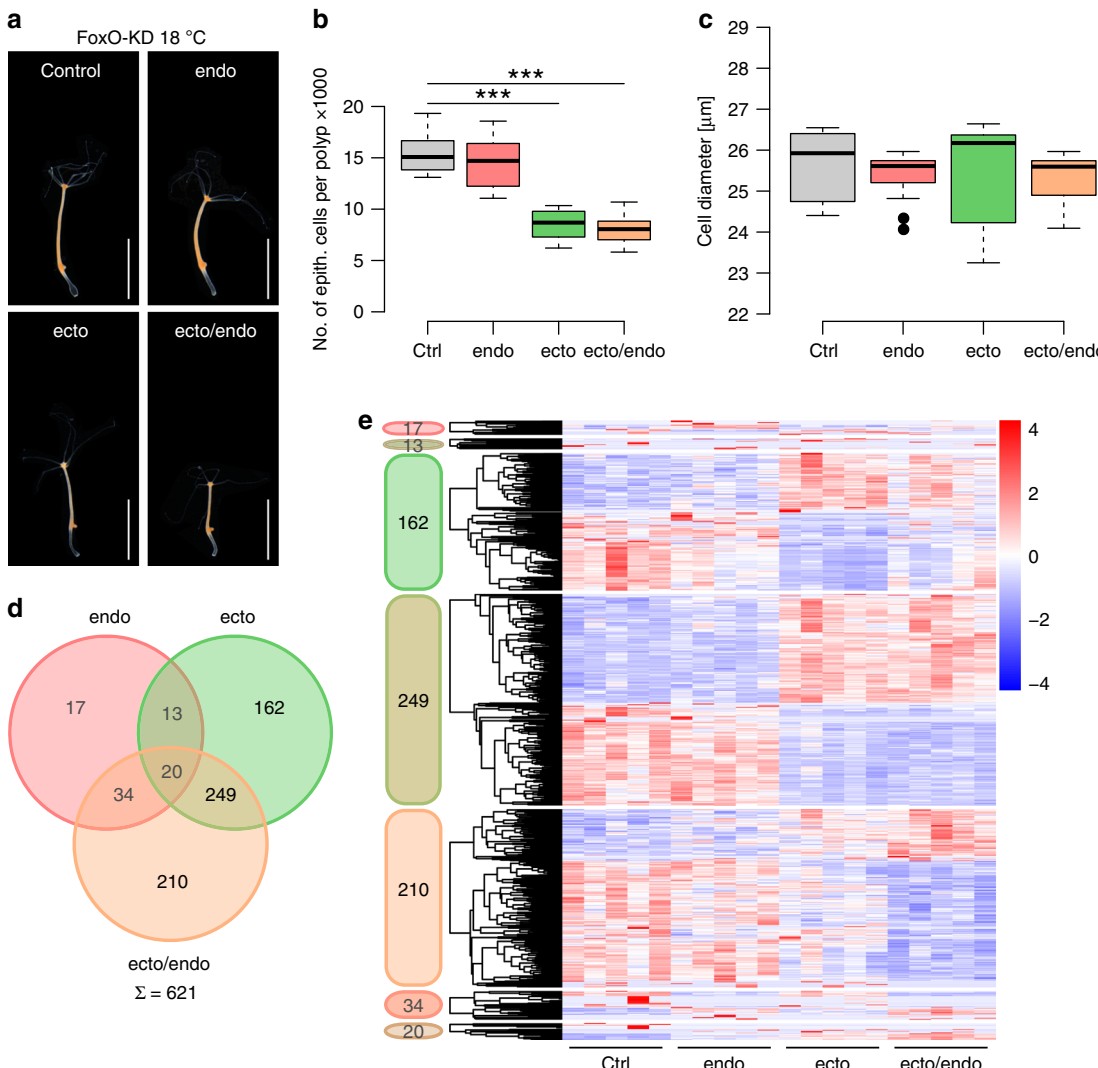

**Fig. 3** FoxO-KD in ectoderm reduces polyp size. **a** Pictures of control and FoxO-KD (endo, ecto, ecto/endo) polyps at 12 °C and 18 °C with bud protrusion. Scale bar: 5 mm. **b** Control and FoxO-KD polyps reared at 18 °C. Animals with an ectodermal FoxO-KD displayed a drastic reduction in size. Endodermal FoxO-KD had no effect on maximum size. **c** The cell size is unaffected by the KD of FoxO. Boxplots show median (horizontal line), upper and lower quantile (box), lower and upper 1.5 times interquartile range (whiskers), and outliers (points). $n = 15$ samples/481830 cells (ctrl), 16 samples/738779 cells (endo), 16 samples/278091 cells (ecto), 16 samples/265796 cells (ecto/endo), $***p \leq 0.001$ ($U$-test + FDR-correction). **d** Venn diagram showing the overlap of differentially expressed genes by FoxO-KD for the different tissues at 18 °C. Twenty contigs are differentially expressed in all KD lines, however, 621 genes (bold) are associated with small size. **e** Heatmaps representing the relative expression level of the differentially expressed genes (rows) from the venn diagram of the FoxO-KD animals. Genes were clustered hierarchically per heatmap. Expression values were $\log_2$-transformed and median-centered by transcript. $n = 5$ libraries (Wald-test + FDR-correction)

animals at 18 °C (Fig. 2d, e) suggesting a more prominent physiological role for the INSR at this temperature. Among the DE genes, one of the gene sets was predicted to be involved in translation (RP-L14, RP-L26e), and one gene, OTOF, is associated with cell membrane vesicle transport in an ion dependent manner. An activin-like protein (ACV) was found among the DE genes, a secreted component of the TGF-β signaling pathway (Supplementary Data 1). This was the only obvious link between temperature-dependent genes and size regulation via INSR.

**FoxO-knockdown in the ectoderm reduces polyp size**. To further investigate the signaling pathways involved in intrinsic size determination, we employed a similar shRNA approach for the single *foxo* gene present in *Hydra* (Supplementary Fig. 4d, e)[35]. In

vertebrates, FoxO is a downstream transcription factor of the INSR, and a major hub in controlling tissue homeostasis and stem cell maintenance[36]. Downregulation of the *foxO* transcript in *Hydra* resulted in a significantly smaller polyp size (Fig. 3a) and lower numbers of epithelial cells per polyp (Fig. 3b). Both ectodermal FoxO-KD lines (ecto and ecto/endo) were smaller in maximum size by more than 41% at 18 °C compared to the control. As in the INSR-KD polyps, we found no evidence for cell size changes due to FoxO-KD (Fig. 3c). The opposing effect of FoxO-KD compared to the INSR-KD in size regulation indicates that in *Hydra* FoxO is negatively regulated by the insulin signaling. This is in concordance with studies in various other organisms[36]. However, at this point we have no further evidences, that insulin signaling and FoxO interact directly in *Hydra*. The effect of the FoxO-KD is cell-lineage specific, since knocking down FoxO in endodermal epithelial cells caused no change in

**Table 1 Regulation of putative candidate genes and corresponding KEGG numbers for size regulation in polyps with INSR-KD or FoxO-KD and after temperature shift**

| | Gene | KEGG No. | Temperature | | INSR-KD | FoxO-KD | | |
| | | | 12 °C vs. 18 °C | 18 °C vs. 22 °C | ecto/endo 18 °C | ecto/endo 12 °C | ecto 18 °C | ecto/endo 18 °C |
|---|---|---|---|---|---|---|---|---|
| TGF-β | ACV (INHB) | K04667 | ↓ | ↓ | ↓ | | ↓ | ↓ |
| | ACVR2A | K04670 | ↑ | | | | | |
| | Cer-1 | K01645 | | ↓ | | | ↓ | ↓ |
| | DAN (GREM) | K19558 | | | | ↑ | | |
| | DAN (NBL1) | K19558 | ↑ | ↑ | | ↑ | | |
| | TGF-β2 | K13376 | ↑ | ↑ | | | ↓ | ↓ |
| | THBS1 | K16857 | ↑ | ↑ | | ↓ | ↓ | |
| Wnt | Wnt11 | K01384 | | | | ↑ | ↑ | ↑ |
| | FZD1/7 | K02432 | | | | ↓ | ↓ | ↓ |
| | Notum | K19882 | ↓ | ↓ | | | | |
| Others | CALM | K02183 | ↑ | ↓ | | | ↑ | |
| | Casp3 | K02187 | ↑ | ↓ | | ↑ | | |
| | Casp7 | K04397 | ↑ | | | | | |
| | FGF-C | K04358 | | | | ↑ | ↑ | ↑ |
| | FGFR1 | K04362 | ↑ | ↑ | | | | |
| | FLNA | K04437 | | | | ↑ | ↑ | ↑ |
| | Notch | K02599 | ↓ | ↓ | | ↓ | | ↑ |
| | K-ras | K07827 | | ↓ | | ↓ | ↓ | ↓ |
| | R-ras | K07829 | ↓ | ↓ | | ↑ | | |

body size (Fig. 3b) and size determination therefore seems to reside within the ectodermal epithelial cells. This observation is consistent with previous studies using natural size mutants and transplantation experiments[37]. As observed for INSR-KD animals, FoxO-KD polyps showed no change in cell proliferation rates (Supplementary Fig. 2c–e) and were still able to increase in maximum size (Supplementary Fig. 5c, d) when reared at 12 °C, which supports the hypothesis that temperature-induced phenotypic plasticity in maximum polyp size may not be directly regulated by FoxO signaling.

In the corresponding transcriptome study, we found a total of 705 DE contigs in the FoxO-KD animals (Fig. 3d, e). The ectodermal KD (444) alters the expression of five times more genes than the endodermal KD (84). Moreover, in both transgenic lines 20% (17/84) and 36% (162/444) of DE genes are regulated in a tissue-specific context (Fig. 3d). This corroborates the observation of different functions for FoxO in the ectoderm and endoderm. In total we found 621 shared DE genes to be associated with reduced polyp size at 18 °C (Fig. 3d, bold) and 517 at 12 °C (Supplementary Fig. 7, italic). Overall, 257 contigs were shared between both temperatures, resulting in 881 contigs that appear specifically associated with a FoxO-dependent small size in *Hydra* (Supplementary Fig. 7). Genes differentially expressed in FoxO-KD animals cultured at 12 °C and 18 °C include: K-ras, two fibroblastic growth factor C (FGFC) like contigs, the actin binding protein Filamin A (FLNA), a disintegrin and metalloproteinase 9 (ADAM9), and Deoxyribonuclease I (DnaseI). Remarkably, the expression of genes involved in axial patterning, Wnt11 and its receptor Frizzled1/7 (Fzd1/7)[19], was also altered by ectodermal FoxO-KD at both temperatures. While Wnt11 expression was increased, Fzd1/7 was downregulated in all samples (Table 1). Moreover, knocking down FoxO in the ectoderm resulted in differential expression of several members of the TGF-β pathway, which was shown previously to be directly affected by Wnt signaling in *Hydra*[38]. Specifically, we observed Thrombospondin 1 (THBS1) to be downregulated in polyps of smaller size at both temperatures (ecto 18 °C and ecto/endo 12 °C). Thrombospondins have been associated with functions in secretion and receptor binding of TGF-β superfamily members[30]. We also observed DAN, an

inhibitor of TGF-β signaling, to be affected by the FoxO-KD only in animals cultured at 12 °C. In contrast, Activin (ACV), Cerberus 1 (Cer-1) and TGF-β 2 were differentially expressed at 18 °C (Table 1). Cer-1 is known for its dual inhibitory function in TGF-β, as well as Wnt signaling[39].

The enrichment of DE genes for extracellular components of TGF-β signaling in the FoxO-KD animals clearly points to a role of this pathway in size regulation in *Hydra*. While our study did not identify a single gene which appears solely responsible for the size differences induced by temperature, INSR-KD and FoxO-KD (Supplementary Data 2), we consistently encountered genes of the TGF-β signaling pathway (Table 1), as well as the Wnt signaling components Wnt11, Fzd1/7 and Notum as putative candidates for a regulatory function. To further investigate the dependence of Wnt components on FoxO and insulin signaling we performed an expression analysis of selected Wnt pathway components via qRT-PCR (Supplementary Fig. 8). We confirmed the Fzd1/7 and Wnt11 expression to be dependent on FoxO and further found previously undetected Wnt3a dependence on FoxO signaling, as well as marginal significant Wnt11 and significant Wnt8 dependence on INSR expression. This confirms the placement of Wnt signaling downstream of the insulin and FoxO pathway.

**Wnt signaling as control board of size determination.** Inspired by our transcriptomic data (Figs. 1–3, Table 1), we suspected developmental programs might mediate the temperature-induced phenotypic plasticity and/or intrinsic size regulation without affecting the cell turnover. The Wnt signaling pathway regulates axial patterning and is associated with head and bud formation in *Hydra*[19]. Following a model of head activation and head inhibition potential[40], we thought of budding as an increased potential to form a second head and thus a second axis in the animal. Since bud formation terminates linear polyp growth, we explored experimentally whether the head activation/inhibition ratio determined maximum polyp size. In order to interfere with Wnt signaling and thus the head activation potential, we exposed polyps to Alsterpaullone (ALP), a GSK3-β inhibitor which induces supernumerary tentacles along the body column[19]. Applying flow cytometry to individual polyps allowed us to correlate the number of ectopic tentacles, the number of epithelial

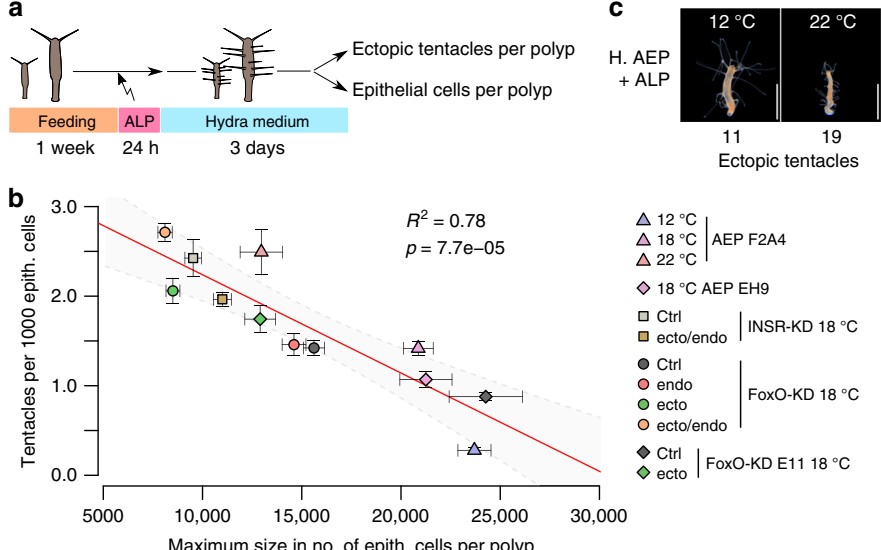

**Fig. 4** Axis stability as a readout for maximum size. **a** Schematic representation of experimental design: After one week of feeding, polyps were treated with ALP. After three days of incubation in HM, ectopic tentacles were counted manually and current size of the same animal was determined using flow cytometry. **b** Scatter plot showing a negative correlation between the maximum size of a *Hydra* line and the responsiveness to ALP treatment. Smaller lines produced more ectopic tentacles per epithelial cells than larger lines. The linear regression line is shown in red, the 95% confidence interval in light gray. $R^2$ represents the coefficient of correlation. Points and whiskers show mean values and standard error. $n = 15$ polyps/18 samples (12 °C F2A4), 6 polyps/18 samples (18 °C F2A4), 8 polyps/11 samples (22 °C F2A4), 12 polyps/12 samples (18 °C EH9), 19 polyps/20 samples (INSR-KD ctrl), 20 polyps/20 samples (INSR-KD ecto/endo), 10 polyps/15 samples (FoxO-KD D11a ctrl), 15 polyps/16 samples (FoxO-KD D11a endo), 8 polyps/16 samples (FoxO-KD D11a ecto), 6 polyps/16 samples (FoxO-KD D11a ecto/endo), 13 polyps/5 samples (FoxO-KD E11 ctrl), 17 polyps/5 samples (FoxO-KD E11 ecto) (tentacles per 1000 epith. cells/maximum size, ANOVA). **c** Clonal animals reared at 12 °C or 22 °C four days after ALP treatment. Note that smaller animals at 22 °C respond stronger to ALP than 12 °C polyps. Scale bar: 2 mm

cells after ALP treatment and the maximum size to investigate the influence of axis stability on maximum size (Fig. 4a, Supplementary Fig. 9). The experiment provided us with three important results: first, an expected significant positive correlation of the individual size and the maximum size of the different polyp strains, which explained around 50% of the variance in the maximum size (Supplementary Fig. 9e). Second, we found a marginal correlation of tentacle number per polyp and maximum size of the line (Supplementary Fig. 9d). Third, we noticed that larger individuals tended to develop more ectopic tentacles in response to ALP (Fig. 4c, Supplementary Fig. 9g) and concluded that the potential of ectopic tentacles generation increases with larger body columns before a bud emerges, simply because of more tissue availability. By correcting the tentacle number per polyp with its individual size we could show a significant correlation between the tentacle number per epithelial cell and the maximum size of the *Hydra* line with high confidence (Fig. 4b; $R^2 = 0.78$, $p = 7.7e{-}05$, ANOVA). The corrected tentacle number explained size regulation at three temperatures and in different genetic backgrounds including all KD lines. These findings demonstrate a negative correlation between maximum size and axis stability and suggest that these two factors are mutually linked. Consequently, differences in body size can be explained by an altered head activation to inhibition ratio and might be directly controlled by Wnt signaling as a measuring tool for the size of the polyp. The size of a polyp before emergence of the first bud protrusion may be defined as critical size and demarcates the change in Wnt signaling to induce a secondary body axis[4].

**TGF-β signaling as effector of size determination.** Since Wnt signaling and size regulation are linked and our transcriptome analysis suggests an involvement of TGF-β signaling (Table 1), we tested whether Wnt signaling controls the TGF-β pathway.

Treating polyps with ALP causes dysregulation of TGF-β pathway components (Supplementary Fig. 10a), placing TGF-β signaling downstream of the Wnt pathway. Thus, we hypothesized that Wnt signaling determines the critical size, while bud initiation is controlled by TGF-β signaling[38]. Using chemical interference, we observed that two specific TGF-β receptor inhibitors, K02288 and LDN-193189[41,42], affected maximum polyp size in a concentration dependent manner (Fig. 5a, b). To assure specificity of the inhibitors in *Hydra* we performed careful sequence based in silico analysis and revealed slight changes in the ATP binding pockets of *Hydra* TGF-β receptors compared to human homologs, which suggest a differential inhibition resulting in LDN-193189 targeting ACVR1 and K02288 targeting TGFR1 most effectively (Supplementary Fig. 11). These different target specificities (Supplementary Fig. 11, Supplementary Table 3[42]) for the TGF-β pathway receptors resulted in opposite effects: While K02288 reduced the number of epithelial cells in the wild type F2A4 by 63% ($25,127 \pm 4172$; $n = 36$; to $9330 \pm 2667$; $n = 12$) at the highest concentration (2 μM) (Fig. 5a), LDN-193189 increased the number of epithelial cells by 49% ($21,306 \pm 4159$; $n = 28$; to $31,702 \pm 9816$; $n = 16$) at the highest concentration (3 μM) (Fig. 5b). In order to confirm the importance of the TGF-β pathway in the signaling hierarchy of molecular mechanisms controlling size, we treated the large-sized INSR-KD animals with K02288 and were able to reduce the polyp size by 23% ($13,928 \pm 1518$; $n = 8$; to $10,755 \pm 1407$; $n = 9$) at 1 μM, which is even smaller than the untreated corresponding INSR-KD control animals (Fig. 5c, Supplementary Fig. 12). In addition, we applied LDN-193189 to small-sized FoxO-KD animals and found an increase of epithelial cells per polyp by 15% (ecto) and 33% (ecto/endo), respectively, at 3 μM (Fig. 5d, e). The potential to alter maximum size in insulin and FoxO signaling-deficient animals with TGF-β inhibitors verifies the hierarchical action of the TGF-β-pathway downstream of insulin signaling and FoxO. We found

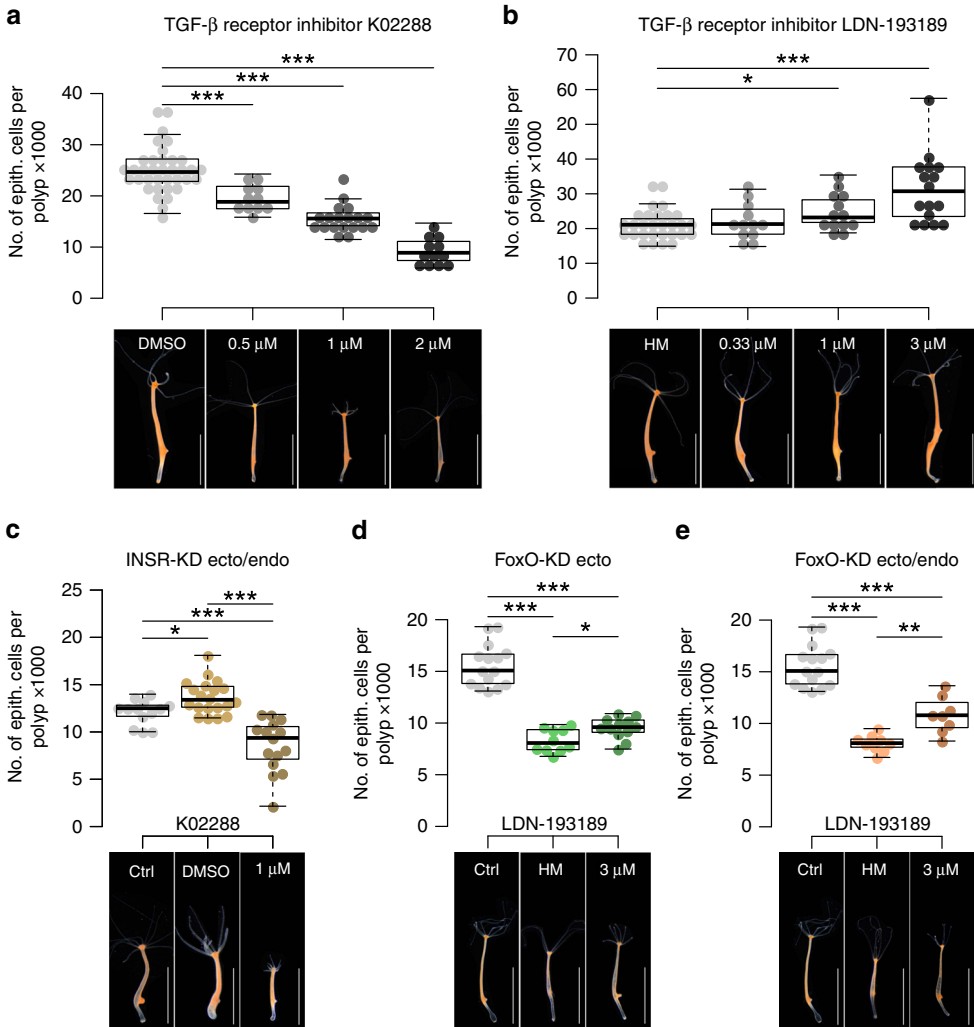

**Fig. 5** TGF-β signaling affects maximum size. Applying TGF-β receptor inhibitors with different specificities led to alteration in maximum size. While K02288 reduced size (**a**), LDN-193189 increased size (**b**) in a concentration dependent manner. Boxplots show median (horizontal line), upper and lower quantile (box), lower and upper 1.5 times interquartile range (whiskers), and outliers (points). $n = 36$ samples (DMSO), 12 samples (K02288, 0.5 μM, 1 μM, 2 μM), 28 samples (HM), 12 samples (LDN-193189 0.33 μM), 8 samples (LDN-193189 1 μM), 16 samples (LDN-193189 3 μM). **c** Large-sized INSR-KD animals were reduced in maximum size by K02288. ctrl = INSR-KD ctrl + DMSO ($n = 13$ samples), DMSO = INSR-KD ecto/endo + DMSO ($n = 20$ samples), 1 μM = INSR-KD ecto/endo + 1 μM K02288 ($n = 15$ samples). **d, e** Small-sized FoxO-KD (ecto and ecto/endo) animals increased in maximum size by LDN-193189. ctrl = FoxO-KD ctrl + HM ($n = 15$), HM = FoxO-KD [ecto|ecto/endo] + HM ($n = 10$ samples/9 samples), 3 μM = FoxO-KD [ecto|ecto/endo] + 3 μM LDN-193189 ($n = 12$ samples/8 samples). Scale bar: 5 mm. $n \geq 8$, *$p \leq 0.05$, **$p \leq 0.01$, ***$p \leq 0.001$, U-test + FDR-correction

no consistent change in cell size in either of the tested conditions after the treatment of *Hydra* polyps with the two TGF-β receptor inhibitors (Supplementary Fig. 13). However, we could not rescue the FoxO-KD to the level of its control line by the application of the TGF-β inhibitors, which could indicate that the TGF-β signal inhibition is not fully achieved at the given concentration of 3 μM LDN-193189. FoxO-KD animals displayed a toxic phenotype at higher concentrations of LDN-193189, probably because FoxO signaling as the most important integrator of environmental stressors[43] was compromised. However, we concluded that the TGF-β signaling pathway controls cell number, but not cell size in *Hydra*. Finally, applying TGF-β receptor inhibitors resulted in a significantly altered response to ALP than supported by the correlation's 95% confidence interval (Supplementary Fig. 10b), suggesting action of TGF-β signaling downstream of Wnt-dependent signaling.

**Maximum size is dependent on developmental time.** Due to their asexual mode of reproduction, *Hydra* polyps continuously

produce clonal offspring. However, parental polyps can only develop buds after they have reached their maximum size. For a *Hydra* polyp, bud initiation determines the end of the developmental growth phase (here termed $\Delta T_1$) since cells gained from proliferation do not contribute to larger size but are displaced to bud development. Consequently, by using the TGF-β-receptor inhibitors K02288 and LDN-193189, we tested experimentally whether maximum size and developmental time ($\Delta T_1$) are directly affected by bud initiation (Fig. 5a, b; Fig. 6a). K02288 indeed decreased $\Delta T_1$ significantly by 23% (8.1 ± 0.8d; $n = 29$; to 6.2 ± 0.7d; $n = 27$) and thereby induced bud initiation earlier in development leading to less epithelial cells per polyp (Fig. 6b). In contrast LDN-193189 significantly increased $\Delta T_1$ by 12% (7.9 ± 0.7d; $n = 29$; to 8.9 ± 1.0d; $n = 29$) explaining larger maximum polyp size (Fig. 6c). Thus maximum polyp size is dependent on the developmental time, where shorter $\Delta T_1$ accounts for smaller maximum size and large animals result from bud suppression and an extended $\Delta T_1$ (Fig. 6a). This process seems to be controlled by the head inhibition potential in the body column as smaller lines

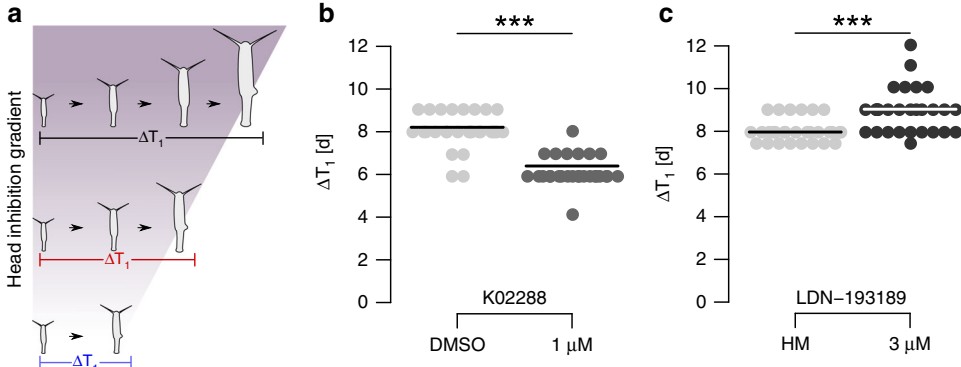

**Fig. 6** Time of bud initiation changes developmental time. Changing developmental time by application of TGF-β receptor inhibitors. **a** Schematic representation correlating maximum size and head inhibition gradient. Larger *Hydra* lines have a higher head inhibition potential, thereby respond less to ALP, and extend in developmental time ($\Delta T_1$). **b** K02288 antedated the time point of bud initiation in developmental time, eventually resulting in smaller maximum size. Points show single samples, line represents the mean. $n = 29$ polyps (DMSO), 27 polyps (K02288). **c** LDN-193189 suppressed bud initiation, leading to longer $\Delta T_1$ and larger maximum size. $n = 29$, ***$p \leq 0.001$, *U*-test

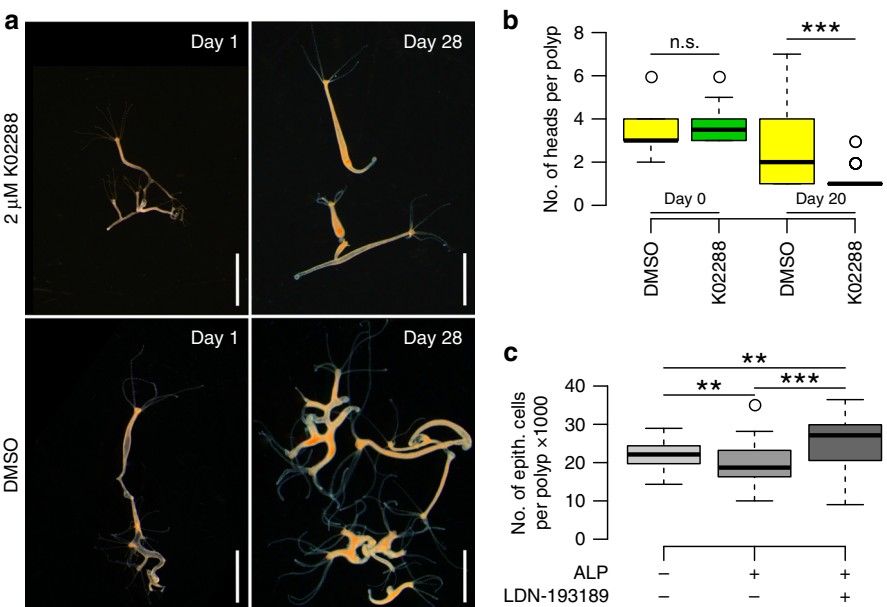

**Fig. 7** Wnt controls TGF-β signaling. β-catenin OE animals exhibit constitutively activated Wnt signaling, leading to the formation of multiple heads and axes. **a** Multiple head phenotype can be rescued with a 28 day treatment of 2 μM K02288 leading to animals with a single axis and one head. Scale bar = 2 mm. **b** Treatment of β-catenin OE *Hydra* polyps reduces the number of heads per polyp compared to DMSO treated controls. Boxplots show median (horizontal line), upper and lower quantile (box), lower and upper 1.5 times interquartile range (whiskers), and outliers (points). $n = 12$ polyps (day 0), 33 polyps (DMSO, day 20), 70 polyps (K02288, day 20), ***$p \leq 0.001$, *U*-test. **c** Low concentration ALP treatment (10 nM) reduces the number of epithelial cells per polyp, compared to DMSO treated controls. Simultaneous treatment with 2 μM LDN-193189 rescued the ALP induced small phenotype and polyps grew to larger sizes compared to the control treatment. $n = 39$ samples (DMSO), 36 samples (ALP), 22 samples (ALP + LDN193189), **$p \leq 0.01$ ***$p \leq 0.001$, *U*-test + FDR-correction

show a higher disposition to form a secondary axis than larger lines.

**TGF-β signaling is a downstream target of Wnt signaling.** In order to verify the hierarchical action of Wnt and TGF-β signaling, we treated animals with constitutively activated Wnt signaling with 2 μM K02288. These animals carried a β-catenin overexpression (OE) construct with a truncated site of degradation of the β-catenin causing accumulation of β-catenin in the cell, leading to multiple heads/axes in the polyps[44,45]. By continuous treatment of β-catenin OE polyps with K02288, we were

able to rescue the multi-headed phenotype and reestablish a single body axis in the animals over the course of three weeks (Fig. 7a, b, Supplementary Fig. 14). The rescue of the β-catenin induced phenotype by the TGF-β receptor inhibitor K02288 clearly indicates that TGF-β signaling is involved in the axis and thus in the bud formation in *Hydra*.

To consolidate the notion that Wnt signaling controls the size by inducing a bud, we treated *Hydra* polyps with low concentrations of ALP (10 nM) to slightly activate Wnt signaling, but avoiding induction of ectopic tentacles. The low ALP treatment resulted in smaller animals, characterized by less epithelial cells per polyp (Fig. 7c). We treated the animals in a

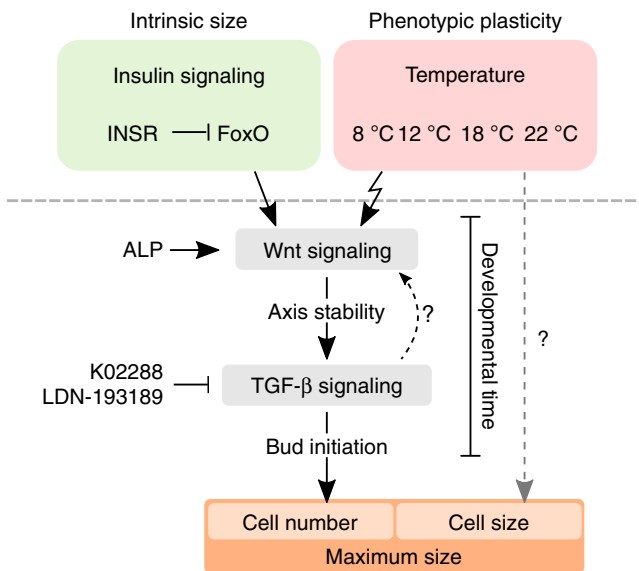

**Fig. 8** Intrinsic size and its phenotypic plasticity are controlled via Wnt and TGF-β signaling. Scheme showing the regulation of intrinsic size and phenotypic plasticity as independent inputs on Wnt signaling. Wnt-dependent axial patterning induces bud initiation via TGF-β signaling, thereby stops the developmental growth phase and determines maximum polyp size. This process can be induced or suppressed using K02288 or LDN-193189, respectively. Cell size regulation is temperature-dependent but independent of Wnt and TGF-β signaling

combination of 10 nM ALP and 2 µM LDN-193189 and could rescue the effect of Wnt-signaling induced reduction in body size leading to the generation of polyps that were larger than the control group (Fig. 7c). These two experiments clearly show that Wnt signaling is involved in the size regulation of *Hydra* and that TGF-β signaling is downstream of the Wnt signaling.

Taken together, by our combined transcriptomic and experimental approach we identified hierarchical action of environmental and genetic factors on size regulation in *Hydra*. We uncovered temperature and insulin signaling as independent inputs on Wnt pathway-dependent axial patterning that in turn controls bud initiation and thereby maximum size via TGF-β signaling. Phenotypic plasticity and intrinsic maximum size determination in *Hydra* are both mediated via developmental time and the onset of bud initiation, with cell proliferation playing no major part (Fig. 8).

## Discussion

Although body size is attributed one of the most important features of all life forms, its regulation at the whole organism level remains poorly understood[4]. Using an integrated approach combining comparative transcriptomics and functional studies in an evolutionary context, we contributed to filling the knowledge gap. We could show that *Hydra's* body size regulation in response to temperature cues and genetic factors such as INSR and FoxO directly affects common developmental mechanisms including the Wnt and the TGF-β signaling pathway.

*Hydra* allows studies of phenotypic plasticity and functional genetic approaches in a single genotype. While previous studies have uncovered Wnt signaling as the key component for axial patterning in *Hydra*[19], we demonstrate that Wnt signaling determines the developmental time until the critical size is reached and break of axis symmetry is initiated. We therefore hypothesize that Wnt-dependent signaling serves as the molecular measuring tool for the tissue to assess its own size. Since

pharmacological interference with TGF-β signaling affected the time of bud initiation and consequently the body size of the animal, TGF-β signaling seems to serve as downstream effector mechanism (Supplementary Fig. 10a) promoting bud initiation[38].

Furthermore, *Hydra* utilizes the insulin signaling, including the FoxO transcription factor, to regulate its own size. FoxO dependent insulin signaling has been recognized for controlling cell proliferation and growth promoting factors[6]. Insulin signaling drives larval growth in model organisms like *Drosophila*[46] and promotes long bone elongation in mammals[45], while interference with the insulin signaling pathway leads to smaller animals in *C. elegans*[47], *Drosophila*[48], and mice[49]. Similarly, the expression of human FoxO showed single nucleotide polymorphisms (SNPs) dependence[45], which seems to be responsible for phenotypes like lifespan extension[50,51] and smaller size[52,53].

We further found that TGF-β is the effector for regulation of cell number in *Hydra*. In *C. elegans*, TGF-β causes changes in cell sizes and controls cell mass directly[54]. In contrast, cell size is regulated temperature-dependently and seems to be TGF-β independent in *Hydra*. In mammals, size is determined by long bone growth and TGF-β signals are needed for bone elongation, as well as bone maturation, and thus size determination[45]. We thus suggest a highly conserved role of insulin and TGF-β signaling for size regulation in all metazoans, though implementation of the exact size regulation mechanism seems organism specific.

*Hydra* size regulation is independent of cell proliferation and mediated by bud initiation. Bud initiation can be considered as a developmental switch between growth and reproductive phase, as it is described for most determinate growing animals[4,6]. In *Hydra*, timing of this switch is determined by a cascade of insulin or FoxO, Wnt, and TGF-β signaling.

Insulin and TGF-β signaling have been described in several model organisms to promote progression through developmental stages. In *C. elegans*, both pathways control formation of the dauer form and are needed to correctly induce molting at appropriate sizes[55], while in *Drosophila* insulin signaling drives pupation at adequate sizes and TGF-β seems to be needed to prime the tissue to receive the insulin signal[6]. Mammals with defects in insulin signaling show delayed sexual maturation[56], and TGF-β signaling is needed for the production of sexual hormones to induce puberty[57,58].

All described model systems seem to measure size and switch developmental programs after passing a critical size threshold[3,4]. Developmental timing depends on reaching these target sizes and can be delayed or shortened depending on environmental conditions. Insulin and TGF-β signaling are needed to promote growth to these developmental brinks and induce the developmental switch. Once the developmental programs have changed, insulin and TGF-β signaling outcomes change as well and drive developmental programs to fix size[3,4]. We show Wnt signaling as mediator between the insulin/FoxO/temperature signaling and the TGF-β pathway and propose a role of Wnt signaling in body size measurement, which in turn functions as reference point to determine developmental timing, at least in *Hydra*. This finding poses the question of whether this form of size measurement is specific to *Hydra* or whether similar mechanisms exist in other organisms. Wnt signaling has been shown to be a genetic contributor to body and organ size in other species, rendering it a good candidate for size measurement among all animals.

Our study shows, that *Hydra* integrates temperature signals into TGF-β signaling. While the mechanism of temperature sensing remains elusive to date, TGF-β signaling has been associated with mainly intrinsic developmental cascades. *C. elegans* releases TGF-β in response to the reception of population density as an environmental factor[59]. Together with our results this

suggests a role of TGF-β signaling as a more general integrator of environmental signals. Finally, the integration of environmental signals into conserved developmental processes highlights the potential of external factors shaping individual development and evolution in all metazoa.

With this study we generated a model, where intrinsic and environmental cues utilize the same developmental pathways to regulate body size in *Hydra*. We suggest, that Wnt signaling serves as size measuring tool for the tissue and determines critical size by control over axis stability. Once the critical size is reached, TGF-β signaling is activated and induces a secondary axis through bud formation, determining maximum size in a cell number dependent manner. Both pathways determine the developmental time for *Hydra* to grow and reach its final body size (Fig. 8). Thus, intrinsic and environmental signals control hierarchical signaling cascade of Wnt and TGF-β signaling, which mediate developmental timing and finally body size.

## Methods

**Animal culture**. Experiments were carried out with *Hydra vulgaris* (strain AEP). All lines were continuously cultured under the described temperature conditions (8 °C, 12 °C, 18 °C or 22 °C) in *Hydra* medium (HM; 0.28 mM CaCl₂, 0.33 mM MgSO₄, 0.5 mM NaHCO₃ and 0.08 mM KCO₃) according to the standard procedure[60]. The animals were fed two times a week at 8 °C and 12 °C and three times a week at 18 °C and 22 °C. During experimental setups the animals were fed four times a week.

**Transgenic animals**. The stable knockdown lines for FoxO[35] or the insulin-like peptide receptor (INSR) were achieved by generating transgenic polyps expressing a FoxO or INSR hairpin construct, respectively, fused to eGFP under control of an actin promoter (Supplementary Fig. 4). The β-catenin overexpression animals carried an actin-promoter driven, truncated (amino acid 1–138 were missing) β-catenin-GFP fusion product[44]. After microinjection of DNA plasmids into early embryos of *Hydra vulgaris* (AEP), we obtained transgenic lines that observe a mosaic pattern after hatching[33]. By continuous selection for eGFP-expression in transgenic cells in the ectodermal and/or endodermal epithelium during asexual reproduction, we created completely transgenic animals for these tissues. At the same time, we also selected for clonal animals without any eGFP expressing cells and thereby created KD line-specific controls[34]. Therefore, each of our *Hydra* KD lines, being INSR-KD or FoxO-KD, comes with its corresponding KD control line that shows the unaffected maximum size for this genotype.

**Tissue digestion and size determination using flow cytometry**. For maximum size determination, animals in the early budding state were selected from the mass culture. For each replicate, individual polyps were digested in 100 µl of 50 U/ml Pronase E (*Serva*) in an isotonic culture medium[61] and inverted every 10 min for 4 h at 18 °C. The living cells were then measured on a BD FACSCalibur with Cell-QuestPro v5.2 (*Becton–Dickinson*) using forward scatter and side scatter with a blue 488 nm laser to gate the epithelial cells. Epithelial cells were previously localized by measuring transgenic GFP expressing epithelial *H. vulgaris* (AEP) lines[33] using the FL-1 filter (530/30 nm) (Supplementary Fig. 1). Gating and further analyses were performed with FCSalyzer 0.9.13-alpha (https://sourceforge.net/projects/fcsalyzer/). Cells of the interstitial cell lineage were not evaluated to exclude effects of germ line producing cells under varying environmental conditions. However, because of their role in determination of morphological features of *Hydra*[37] and since they are much larger than cells from the interstitial cell lineage, epithelial cells are a very good proxy for the maximum polyp size[61,62].

**Transcriptome assembly and annotation**. To improve mapping of low expression genes, a new *Hydra vulgaris* (AEP) transcriptome was assembled from 25 libraries of the control samples. The libraries were corrected for sequencing errors using Rcorrector 1.0.2[63]. Afterwards possible adapter sequences tailing the reads were removed applying Cutadapt 1.13[64] and residual rRNA reads were filtered by mapping the libraries against the SILVA rRNA database[65] using Bowtie 2 2.2.9[66]. The Trinity assembler[22] was used to generate a de-novo transcriptome, performing in silico normalization to keep a maximum of 50 redundant reads and allowing a minimal coverage of five bases for generating confident contigs. To reduce the size of the resulting transcriptome we performed two clustering steps: First we clustered the sequences by similarity applying CD-HIT-EST, allowing 1% divergence[67]. Second we quasi-mapped all 60 libraries to the given transcriptome using sailfish 0.10.0[68] and clustered the sequences for similarity of shared equivalence classes using RapClust 0.1.2[69]. We isolated the longest sequence as the gene representative from the resulting clusters and used the obtained and reduced transcriptome to predict open reading frames (ORFs) utilizing Transdecoder 5.01 (https://github.com/TransDecoder/TransDecoder). The transcriptome contains 31,325 contigs

with a predicted open reading frame (ORF), longer than 100 amino acids and a translation start or stop or both within the sequence range. We used ORF containing contigs for further analysis, and therefore refer to the 31,325 members as genes. The predicted peptides were than submitted to a web based KO-annotation tool BlastKOALA[70]. Furthermore a SMART, Pfam domain and PANTHER family prediction was performed using the InterProScan tool 64.0[71]. Where possible, we obtained a reviewed UniProt member of all annotated PANTHER subfamilies and annotated the KO number of this UniProt member. All transcriptome data are deposited in the Sequencing Read Archive (SRA) and can be found using the accession number SRP133389.

**RNA-Seq analyses**. For transcriptome sequencing transgenic lines were cocultured in shared HM with according controls for at least four weeks in five independent replicates. After sampling, animals were frozen in TRIzol (*Thermo Fisher Scientific*) at −20 °C until RNA-extraction with the PureLink RNA Mini Kit (*Ambion*) according to the manufacturer's protocol. In addition, the optional on-column DNA digestion was performed. The RNA was eluted in 30 µl and checked for sufficient quality. If necessary, the RNA was purified using 1-butanol and diethyl ether[72] and frozen at −80 °C until further use. Total RNA sequencing with previous ribosomal depletion was performed for 60 libraries on the Illumina HiSeq2500 v4 platform, with 125 bp paired-end sequencing of 12 libraries per lane. This resulted in 30–40 million reads per sample after quality control. Quality and adapters were trimmed using PRINSEQ-lite 0.20.4[73] and Cutadapt 1.13[64]. Subsequently, mapping against the newly assembled transcriptome was performed using Bowtie 2.2.9[66]. All downstream analyses were conducted in 'R'. Differentially expressed contigs were identified with the package DESeq2 1.16.1[74] after batch correction with SVA[75].

**TGF-β receptor inhibitor experiments**. To investigate the effect of TGF-β receptor inhibitors on maximum polyp size and the phenotype of β-catenin OE, budless animals were continuously treated with different sublethal concentrations of the inhibitors LDN-193189 (*Sigma-Aldrich*) and K02288 (*Sigma-Aldrich*) at 18 °C. K02288 stock solutions of 10 mM in DMSO were stored at −20 °C and further diluted with HM to the according concentrations. LDN-193189 was dissolved in HM at a stock solution of 12.3 mM and 10 mM, respectively, and also stored at −20 °C until further dilution. After at least one week of incubation with daily feeding, animals with a bud protrusion were collected and digested for flow cytometry analysis and maximum size determination. To show the involvement of developmental time ΔT₁ (bud detachment to first bud protrusion) in the maximum polyp size, overnight detached buds were immediately incubated in the according inhibitor (LDN-193189 3 µM; K02288 1 µM) and starved for five days. Under daily feeding the animals were further cultured in the inhibitors until first bud protrusion. In all experiments where inhibitors were dissolved in DMSO, control medium contained an equal concentration of DMSO (<0.05% (v/v)). To investigate the effect of K02288 on the β-catenin OE animals, single multi-headed *Hydra* polyps were placed into a 12-well plate cavity and permanently incubated with HM containing 2 µM K02288. The number of heads, the number of polyps and the heads per polyp for each well was recorded over the course of the experiment.

**Alsterpaullone experiment**. To examine the effect of axis stability on maximum polyp size, Alsterpaullone (ALP; *Sigma-Aldrich*) was used to inhibit the Glycogen synthase kinase 3 (GSK3) and activate β-catenin dependent gene expression[19]. Budless polyps were incubated for 24 h in 5 µM ALP in HM at the stated temperature, subsequently washed and incubated for three days in HM. On day four of the treatment, ectopic tentacles were counted and polyps were prepared for flow cytometry to determine the current polyp size. Animals cultured at 12 °C were incubated for 48 h in 5 µM ALP to induce formation of ectopic tentacles. The ratio of tentacles per 1000 epithelial cells was calculated for every polyp individually. For the permanent treatment with ALP, animals were reared in HM containing 10 nM ALP at 18 °C for 10–14 days. Animals were fed four times a week throughout the treatment.

**Cell cycle analysis**. In general, cell cycle analysis was performed as previously described using BrdU-labeling[76]. For 72 h BrdU-labeling animals where incubated in HM containing 1 mM BrdU. For analysis of BrdU labeled cells using a cytometer, cells were dissociated as described above and fixed for 10 min in 4% PFA. The cell's DNA was denatured using 1.5 M HCl for 45 min at room temperature, primary antibody (mouse-α-BrdU, *Roche*) was applied 1:100 in PBS + 1% BSA over night at 4 °C. Secondary antibody (donkey-α-mouse 488 alexa fluor coupled, *Invitrogen*) was incubated 1:500 in PBS + 1% BSA for 1 h at room temperature. Samples were analyzed immediately after staining.

**In-silico analysis of TGF-β receptor inhibitor binding**. Human sequences for the analysis were obtained from the Uniprot database with the following identifiers: P37023 (ACVRL1), Q04771 (ACVR1), P36894 (BMPR1A), P36896 (ACVR1B), P36897 (TGFR1), Q8NER5 (ACVR1C), O00238 (BMPR1B), P37173 (TGFR2), Q13873 (BMPR2), P27037 (ACVR2A), and AQ13705 (ACVR2B). *Hydra* TGF-β receptor sequences were identified from the new transcriptomic data. Multiple sequence alignments were performed using the MUSCLE algorithm[77]. For tree

building a nearest neighbor interchange (NNI) maximum likelihood tree applying the LG model[78] with a gamma distribution (5 discrete gamma categories) and invariant sites starting from an initial neighbor joining (BioNJ) tree was calculated. Dayhoff distances were inferred from the initial multiple sequence alignment and the sum of the distances to the human ACVRL1 and ACVR1 sequence was used to build the linear regression models. IC50 values were obtained from the literature[41,42].

**Statistics**. Statistical analyses were performed using two-tailed Student's-*t*-test or Mann-Whitney-*U*-test if applicable. If multiple testing was performed, *p*-values were adjusted using the Benjamini-Hochberg correction[79].

## Data availability
The authors declare that all data supporting the findings of this study are available within the article and its supplementary information files or from the corresponding author upon reasonable request.

The newly generated sequencing data sets of this study have been deposited in the Sequencing Read Archive (SRA) database under the accession code SRP133389.

## Code availability
Experimental datasets and scripts to generate graphs are provided at figshare.com [https://figshare.com/articles/2019_Mortzfeld_et_al_NatCom_RawData_tar_gz/8178830].

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

## Acknowledgements

We greatly appreciate the equipment and expertize for flow cytometry measurements provided by Maren Falk-Paulsen from the Institute of Clinical Molecular Biology (IKMB) Kiel, Germany. This work was supported by the Deutsche Forschungsgemeinschaft (DFG) (CRC1182 'Origin and Function of Metaorganisms', DFG grant BO 848/15-3, and grants from the DFG Cluster of Excellence program 'Inflammation at Interfaces'). T.C.G. B. acknowledges support from the Canadian Institute for Advanced Research (CIFAR).

## Author contributions

B.M.M., J.T., A.V.K., S.F., and T.C.G.B. designed experiments; B.M.M. and J.T. performed experiments; B.M.M., J.T., A.V.K., S.F., and T.C.G.B. analyzed data; P.R. provided RNA sequencing and flow cytometry. B.M.M., J.T., and T.C.G.B. wrote the paper.

## Additional information

**Competing interests:** The authors declare no competing interests.

