## [Peer Review File · Nature Communications]

Reviewers' Comments:

Reviewer #1:

Remarks to the Author:

The authors attempted to investigate the molecular and genetic mechanism in determination of organismal size especially in freshwater polyp Hydra. They found that knockdown of insulin like receptor (INSR) surprisingly increases maximum polyp size and the number of epithelial cell, which phenotype was reversed when FOXO was knock downed. To further investigate the mechanism, they used temperature dependent trait of size differentiation by rearing the hydra in different temperatures and investigated the differentially expressed genes depending on specific temperatures. Many genes involved in Wnt and TGF- β pathways were screened as candidate genes that mediate INSR/FOXO dependent size regulation. By using several inhibitors of Wnt and TGF- β signalling, they propose the INSR-Wnt-TGF β cascade that determines size of the hydra.

I feel that it is an interesting paper that incorporates temperature dependent traits of size differentiation and their use in differential expression analysis to define molecular pathways of size regulation. However, I feel that more works should be needed to sufficiently support the model proposed by the authors (Figure 7) to guarantee the publication in top-tier journal

1. It is a surprising observation that knockdown of INSR actually increases hydra's size. In most of vertebrate and invertebrate animal models, it is generally known that insulin/insulin like signalling positively regulates cellular and organismal growth, so the authors' observation is quite unexpected. As such, I feel that rigorous work should be done to more strongly support this finding. The authors should consider off-target effects when knockdown of INSR using shRNA, so additional independent shRNAs should be used to confirm the reproducibility of the data presented. Moreover, the authors should confirm the activity of molecular marker of insulin signalling when insulin signalling is perturbed.
2. I assume that in Figure 4, the authors tried to show that Wnt signal mediated tentacle formation (axis stability) is negatively correlated with organismal size associated with INSR and FOXO. To argue for the model for INSR-Wnt-TGF β cascades in size determination, the authors should show that size-modified hydra induced by knockdown of INSR or FOXO exhibits correlative tendency of tentacle formations or related traits reflecting axis stability. Present form of Figure 4 does not support that insulin signalling controls hydra's size by regulating Wnt signalling mediated axis stability.
3. Figure 5 showed that treatment of TGF- β inhibitors modifies hydra's size both in wild type and in conditions of perturbation of insulin signalling. In order to establish that insulin signalling controls hydra's size through TGF- β , the authors should show that perturbation of insulin signalling by knockdown of INSR or FOXO barely modulates hydra's size in the presence of TGF- β inhibitors.
4. Figure 5 used two different TGF- β signalling inhibitors which lead to two contradictory results. The authors noted that it may be due to the different target specificities referring to table s3, which fails to sufficiently explain the results. Why do two TGF- β inhibitors having similar target affinities have opposite effects on hydra's size regulation?
5. Figure S8 shows that treatment of GSK-3 inhibitor decreases the expression of several TGF- β signalling component genes, which cannot sufficiently support that TGF- β signalling is downstream of Wnt. To support this, the authors should show that Wnt signalling induced changes in organismal size diminish in the presence of TGF- β inhibitors.
6. In Figure 2d the authors noted that no significant size difference was observed at 8°C, but there are no data showing this.
7. There are numerous errors in numbering of supplementary figures in the text

98line Supplementary Fig S3a -> Supplementary Fig S2a
163line Supplementary Fig S3b -> Supplementary Fig S2b
168line Supplementary Fig S2a,b -> Supplementary Fig S3a,b
217line Supplementary Fig S3c-e -> Supplementary Fig S2c-e
218line Supplementary Fig S2c,d -> Supplementary Fig S3c,d
257line Supplementary Fig S4a -> Supplementary Fig S5a
260-261line Supplementary Fig S4b-d -> Supplementary Fig S5b-d.

Reviewer #2:

Remarks to the Author:

This study addresses an interesting biological question, how genetic and environmental factors do interact to regulate animal size. In their manuscript Mortzfeld et al use the freshwater Hydra polyp as a model and combine transcriptomics, transgenesis and pharmacological approaches to investigate this question. They claim that three distinct signaling pathways, namely the Insulin, Wnt and TGF- β pathways contribute to the regulation of cell number in Hydra. As main result, they propose that Wnt signaling acts as a size-measuring tool and controls axis stability by determining "critical size" at which budding is initiated. Once the critical size is reached, TGF- β would induce secondary axis formation through bud formation. In addition the authors also explore the regulation of cell size that seems to be rather temperature-dependent but independent of Wnt and TGF- β signaling. Unfortunately, the paper lacks coherence and is not convincing in the present state.

Transcriptomics do not suffice to evidence changes in growth factor signaling. Also the knock-down assays of the insulin receptor and FoxO, obtained through shRNA transgenic constructs, are not validated and provide results at the phenotypic and transcriptomic levels that are rather patchy and descriptive. The transcriptomic analyses of these animals only provide a list of putative target genes of insulin signaling, while the pharmacological experiments designed to modulate the activities of the Wnt and TGF- β pathways do not suffice to decipher their role. In addition several technical issues need to be solved to convincingly demonstrate a correlation between temperature and animal size. Therefore, the significance of this article is currently too limited for the broad readership of Nature Communications.

Here are listed the main criticisms that should be addressed by the authors:

1. The authors demonstrate that animals are significantly larger at lower temperatures. However, animal size varies with the culture conditions such as for example the number of animals in a dish in a given volume. We noted some strong variations in the average size of the wild-type (Fig 1b) and control animals (Fig 2b, 3b, Supplementary Figure S3b, d). The authors definitely need to verify throughout their experiments the stability of animal size in the culture conditions they use.
2. The opposing effects of FoxO-KD and INSR-KD in size regulation suggests that Insulin signaling negatively regulates Hydra FoxO. To confirm this results the authors should validate their KD assays and show that FoxO activity is indeed enhanced when INSR is knocked-down.
3. The scale of the plots does not seem consistent between the different experiments. The authors should explain why in all experiments GFP negative control animals that are supposed to be equivalent to the wild-type condition are always smaller. It is quite difficult to evaluate the role of INSR-KD in the regulation of cell number, if at 22°C the control animals are 50% smaller than the wild-type ones.
4. The authors propose that Wnt signaling responds to Insulin signaling independently of the temperature effect (Fig 7). The argument is the up-regulation of Wnt11 in FoxO-KD animals at 12°C and 18°C. However, Wnt11 expression does not seem to be modulated in INSR-KD animals kept at 18°C (Table 1). The expression of several other Wnt ligands should be tested in FoxO-KD and INSR-KD animals. In addition, the authors need to investigate the activity of the Wnt pathway in these different contexts to convincingly establish a hierarchical link between Insulin and Wnt signaling.
5. The authors propose that the activity of the TGF- β pathway is under the control of Wnt, as

proposed by Watanabe et al in 2014 (should be cited for this point). To support this claim, the authors need to provide functional evidences. The results shown in supplemental figure 8 rather suggest that Wnt signaling negatively regulates TGF- β signaling.

6. Figure 4b requires better explanation to adequately evaluate the data. Again, it is difficult to understand why the GFP negative control animals of the different KD lines differ so much in size and tentacle number after ALP treatment. The presented results indicate that these controls are not equivalent to each other and also not equivalent to the AEP F2A4 wild-type animals.

7. The authors show opposite effects of the two TGF- β receptor inhibitors: a reduction in the number of epithelial cells with K02288 and an increase with LDN-193189. However, the specificity of these drugs in Hydra is not tested. Also their respective cellular effects is not tested, as for example a possible increase in apoptosis upon K02288 exposure. In addition, there is no indication on the temperature(s) where these drug treatments were performed. This information is needed given the temperature impact on animal size.

Minor points:

- lines 335 and 338: the authors write that the increase in time before budding explains the increase in size of the KD animals, referring there to Figure 6. This is misleading as Figure 6 only contains information about dT and the graphs representing the animal size are in Figure 5.
- Line 117: Please clarify how the 20 genes were selected.
- In lines 256 and 260 the authors refer to the Supplementary Figure S4 that should represent the fractions of orphan genes in the analysis. This figure does not seem to exist.
- Accession numbers are missing. All raw sequencing data should be made available.
- The references for Supplementary Figure S2 and S3 are inverted in the text.

Reviewer #3:

Remarks to the Author:

Reviewer #3:

Remarks to the Author:

Overall Impression

- Interesting, solid work that makes an important contribution to the literature

Major Issues

- Well written
- Novel work
- Methodologically sound in general but can the authors clarify how they controlled for multiple statistical comparisons that may lead to a Type I error (i.e. false positive findings) ?
- The discussion should try to tie in the human work in this area to make the findings more relevant across species. While the authors briefly mention genetic influence over human body size there is a body of work on FOXO3, body size and longevity in humans that support the manuscript's current findings and could strengthen the conclusions. This deserves to be mentioned in another paragraph or two in the discussion. Some relevant papers appear below.
- **Q He**, BJ Morris, JS Grove, H Petrovitch, W Ross, KH Masaki, B Rodriguez, ...
Shorter Men live Longer...PLoS One 2014: 9 (5), e94385.
- **Bartke A.** Somatic Growth, Aging and Longevity. NPJ Aging Mech Dis. 2017 Sep 29;3:14. doi: 10.1038/s41514-017-0014-y. eCollection 2017
- **Davy P et al.** FOXO3 and Exceptional Longevity: Insights From Hydra to Humans. Current topics in developmental biology 2018:127, 193-212.
 - o
- **Samaras TT.** How height is related to our health and longevity. Nutr Health. 2012 Oct;21(4):247-61. doi: 10.1177/0260106013510996. Review.
- **Nebel and Bosche.** Evolution of human Longevity. Lessons from Hydra. Aging (Albany NY). 2012 Nov;4(11):730-1.

Minor Issues

- Line 319: the word "neither" should be "either."

Our responses and changes to the manuscript are given in **bold**.

Reviewer #1 (Remarks to the Author):

1. It is a surprising observation that knockdown of INSR actually increases hydra's size. In most of vertebrate and invertebrate animal models, it is generally known that insulin/insulin like signalling positively regulates cellular and organismal growth, so the authors' observation is quite unexpected. As such, I feel that rigorous work should be done to more strongly support this finding. The authors should consider off-target effects when knockdown of INSR using shRNA, so additional independent shRNAs should be used to confirm the reproducibility of the data presented. Moreover, the authors should confirm the activity of molecular marker of insulin signalling when insulin signalling is perturbed.

Thank you for the comment, we agree that an increase of body size by the KD of the INSR was a surprising finding. However, KD of the INSR caused only mild changes in body sizes. To account for off-target effects, we performed a BLAST search for the *insR*-hairpin (HP) construct using an e-value threshold of 10 and no other major criteria to obtain all possible off-targets present in the new *Hydra* transcriptome and checked their expression levels from our RNA-seq experiment (Supplementary Fig. S5c). We could not detect any downregulation in any of the potential target genes except for the INSR itself. Hence, we have no reason to believe that off-target effect for this construct exist. Furthermore, we generated another line bearing the same HP-construct which showed a comparable large size phenotype and added these data in Supplementary Fig. S7a.

Unfortunately at the time of the experiment we were unaware of any definite molecular markers for the insulin signaling in *Hydra* and thus were unable to check the activity of those. In fact, part of our study was to identify factors, via an RNA-Seq experiment, that are controlled by the insulin signaling in *Hydra*.

However, to consolidate the finding of larger polyps by the knockdown of the INSR, we measured the growth phase (ΔT_1) in these polyps and found a prolonged time of growth in INSR-KD polyps (Supplementary Fig. S7b, c), explaining the gain in cell number in these polyps. We added this result in the manuscript in lines 166-171. Thus insulin signaling is not inducing cell proliferation as in other model organisms (Supplementary Fig. S2) but controls developmental decisions in *Hydra*. There is a similar large size phenotype in *D. melanogaster* if interference with the insulin signaling occurs specifically in the prothoracic gland¹. By this interferences, developmental decisions are delayed and the flies reach larger sizes. The ancient role of the insulin signaling might be involved in developmental decision making and not only acting as growth factor, what it is mainly recognized for in the current literature.

2. I assume that in Figure 4, the authors tried to show that Wnt signal mediated tentacle formation (axis stability) is negatively correlated with organismal size associated with INSR and FOXO. To argue for the model for INSR-Wnt-TGF β cascades in size determination, the authors should show that size-modified hydra induced by knockdown of INSR or FOXO exhibits correlative tendency of tentacle formations or related traits reflecting axis stability. Present form of Figure 4 does not support that insulin signalling controls hydra's size by regulating Wnt signalling mediated axis stability.

Thank you for this comment. In order to give a better explanation of the dataset and to convey the intention of the plot in a clearer way, we split Figure 4 into several subfigures and added Supplementary Fig. S10 to the manuscript. Indeed, we saw a correlation of Wnt signal mediated tentacle formation (axis stability) and the maximum size in different wild type genotypes (blue to magenta triangles, and diamond), animals reared at different temperatures (triangles), knockdown of the INSR (squares), and the knockdown of FoxO (circles, diamonds). The plot thus clearly shows, that the Wnt signaling by the means of generation of ectopic tentacles is altered in the INSR-KD or FoxO-KD *Hydra* lines. However, their number of ectopic tentacles is still proportionate to the maximum size and fits to the regression line. Therefore, we can show that Wnt signaling is altered with the interference of the insulin and the FoxO signaling pathway and that Wnt is the key regulator for controlling the maximum size in the polyp. Taken together, we introduce first evidence of the hierarchy of Insulin/FoxO/Temperature – Wnt is controlling size. We edited text in lines 294-309 of the manuscript and added a Supplementary Fig. S10 including an explanatory caption.

3. Figure 5 showed that treatment of TGF- β inhibitors modifies hydra's size both in wild type and in conditions of perturbation of insulin signalling. In order to establish that insulin signalling controls hydra's size through TGF- β , the authors should show that perturbation of insulin signalling by knockdown of INSR or FOXO barely modulates hydra's size in the presence of TGF- β inhibitors.

Thank you for this recommendation! In order to put the results better into perspective, we added controls in the plots of Fig. 5, edited the captions and edited text in lines 341-342 and lines 349-354 of the manuscript.

4. Figure 5 used two different TGF- β signalling inhibitors which lead to two contradictory results. The authors noted that it may be due to the different target specificities referring to table s3, which fails to sufficiently explain the results. Why do two TGF- β inhibitors having similar target affinities have opposite effects on hydra's size regulation?

Thank you for this important question. At first we were also puzzled by this observation and could not explain these surprising results. In order to get a better understanding of the mode of action of the two inhibitors, we performed a detailed analysis of the ATP-binding domain of human and *Hydra* TGF- β receptors. We were able to confidently predicted similar target specificities of both inhibitors for the human TGF- β receptors, but also found differential affinities for the *Hydra* TGF- β receptor ATP-binding domains of the two inhibitors. This result on the one hand suggests activity of the inhibitors in *Hydra* and on the other hand gives an explanation of different effects of the inhibitors, as the inhibitors block different TGF- β receptors and thus different parts of the pathway. Uncovering the exact mechanism and how budding is induced by the interplay of TGF- β pathway components is beyond the scope of this paper but will be worthwhile to look into in the near future as it might reveal conserved interactions between the different parts of the TGF- β pathway. We added these results in lines 329-333 and provided an additional supplementary figure S12.

5. Figure S8 shows that treatment of GSK-3 inhibitor decreases the expression of several TGF- β signalling component genes, which cannot sufficiently support that TGF- β signalling is downstream of Wnt. To support this, the authors should shows that Wnt signalling induced changes in organismal size diminish in the presence of TGF- β inhibitors.

Thank you for this critical comment and the suggestion for further experiments. We extended the evidence, that place the Wnt signaling cascade upstream of the TGF- β pathway by the following experiments. First, we treated β -catenin overexpression animals with K02288. The overexpressed β -catenin in these animals was truncated and lacks the amino acid which is usually phosphorylated by the APC for proteolytic degradation in the absence of a Wnt signal. The blocked degradation of the β -catenin led to the constitutive activation of the Wnt signaling pathway and the formation of multiple axis². Treatment with K02288 rescued this phenotype and restored a single axis in these animals. Second: we treated animals with low concentrations of ALP over a period of 10-14 days and observed a reduction in the number of epithelial cells, confirming our notion of Wnt being the key regulator in size determination in *Hydra*. We next co-treated animals with ALP and LDN-193189 to overwrite the ALP effect with the TGF- β inhibitor. We were able to rescue the small size phenotype in these animals and even observed an increase in epithelial cell number. We included these results in two additional paragraphs in lines 376-395 and added two new figures (Fig. 7, original Fig. 7 is now Fig. 8 and Supplementary Fig. S14) to the manuscript.

6. In Figure 2d the authors noted that no significant size difference was observed at 8°C , but there are no data showing this.

Thank your for this comment, there is data showing this, in the supplementary figures. We added a reference to Supplementary Fig. S6 in the text (Line 184) and the caption of Fig. 2, which shows the temperature effect on INSR-KD and FoxO-KD animals.

7. There are numerous errors in numbering of supplementary figures in the text

8line Supplementary Fig S3a -> Supplementary Fig S2a

163line Supplementary Fig S3b -> Supplementary Fig S2b

168line Supplementary Fig S2a,b -> Supplementary Fig S3a,b

217line Supplementary Fig S3c-e -> Supplementary Fig S2c-e

218line Supplementary Fig S2c,d -> Supplementary Fig S3c,d
257line Supplementary Fig S4a -> Supplementary Fig S5a
260-261line Supplementary Fig S4b-d -> Supplementary Fig S5b-d.

We are sorry for the inconvenience and corrected references to the supplementary figures.

References:

1. Mirth, C., Truman, J. W. & Riddiford, L. M. The Role of the Prothoracic Gland in Determining Critical Weight for Metamorphosis in *Drosophila melanogaster*. *Curr Biol* **15**, 1796–1807 (2005).
2. Gee, L. *et al.* β -catenin plays a central role in setting up the head organizer in hydra. *Dev Biol* **340**, 116–124 (2010).

Reviewer #2 (Remarks to the Author):

1. The authors demonstrate that animals are significantly larger at lower temperatures. However, animal size varies with the culture conditions such as for example the number of animals in a dish in a given volume. We noted some strong variations in the average size of the wild-type (Fig 1b) and control animals (Fig 2b, 3b, Supplementary Figure S3b, d). The authors definitely need to verify throughout their experiments the stability of animal size in the culture conditions they use.

Thank you for this important comment. While it is correct that the culture conditions can affect the size regulation of the animals, we want to highlight that culture conditions were comparable between all treatments, KD animals, and control animals as well as between all lines. Differences between the control animals can be explained due to genetic variation between the lines. Transgenic animals are generated by DNA construct injection into the 1-4 cell stage of the animals, and thus requires sexual reproduction of the polyps. Transgenic hatchlings are usually mosaic, and fully transgenic animals are generated by continuous bud selection resulting in animals containing only transgenic cells¹. The same technique is applied to generate control animals of the same line (same genetic background), by simply selecting animals which do not contain transgenic cells². This explains the differences in the control lines between different experiments and is an effect of different genetic backgrounds, rather than uncontrolled culture conditions.

2. The opposing effects of FoxO-KD and INSR-KD in size regulation suggests that Insulin signaling negatively regulates Hydra FoxO. To confirm this results the authors should validate their KD assays and show that FoxO activity is indeed enhanced when INSR is knocked-down.

Thank you for this comment and we are happy to discuss this point. We are aware that our experimental setup and the collected data are not able to show direct interactions between the insulin receptor and its putative downstream transcription factor FoxO in *Hydra*. However, in independent experiments, we could show that downregulation of *foxO* and the *insR* result in newly discovered size phenotypes. Our manuscript discusses the mechanisms of size determination in basal metazoans and describes intrinsic (genetic) and extrinsic (environmental) factors as independent inputs on the conserved size machinery. Thus, the underinvestigated direct or indirect interaction between FoxO and the INSR is not crucial for this manuscript, since we only acknowledge the observation of different size phenotypes and focused our research on their molecular understanding. However, since the inverse phenotypes for the two KDs are consistent with the literature for other model organisms, we decided to visualize the current understanding of this pathway in our model. Nevertheless, we are grateful for your comment and changed our phrasing slightly to distinguish between Insulin and FoxO signaling and their effect on body size throughout the paper. We also added a sentence stating the lack of knowledge in *Hydra* in line 215-216.

3. The scale of the plots does not seem consistent between the different experiments. The authors should explain why in all experiments GFP negative control animals that are supposed to be equivalent to the wild-type condition are always smaller. It is quite difficult to evaluate the role of INSR-KD in the regulation of cell number, if at 22°C the control animals are 50% smaller than the wild-type ones.

Thank you for this comment. We again would like to point out, that different lines, especially those gained through transgenesis, are the product of sexual reproduction. It is true that lines

with different genetic backgrounds exhibit different sizes. However, it was by chance that the two intensively studied transgenic lines were smaller than the wild type counterpart. In order to show evidence for this claim, we included data for an additional INSR-KD (C6-2) and an additional FoxO-KD line (E11) into our data set. Both transgenic control lines show similar maximum sizes as the wild type F2A4 line (Supplementary Fig. S7a, Supplementary Fig. S10a), thus the smaller maximum sizes of INSR-KD C1-1 ctrl and FoxO-KD D11a ctrl animals are not caused by the transgenesis of the animals.

4. The authors propose that Wnt signaling responds to Insulin signaling independently of the temperature effect (Fig 7). The argument is the up-regulation of Wnt11 in FoxO-KD animals at 12°C and 18°C. However, Wnt11 expression does not seem to be modulated in INSR-KD animals kept at 18°C (Table 1). The expression of several other Wnt ligands should be tested in FoxO-KD and INSR-KD animals. In addition, the authors need to investigate the activity of the Wnt pathway in these different contexts to convincingly establish a hierarchical link between Insulin and Wnt signaling.

Thank you for this criticism and the suggestion for further experiments to support our data set. In new experiments, we tested the expression of several Wnt pathway components via qRT-PCR in the INSR-KD and FoxO-KD animals to confirm the notion of insulin and FoxO dependent Wnt signaling (Supplementary Fig. S9). Indeed, we revealed Wnt11 to be most probably INSR dependent ($p = 0.09$) using this more sensitive method, while we also identified Wnt8 to be insulin signaling dependent. We further confirmed the FoxO dependent Fzd1/7 and Wnt11 expression of the transcriptome analysis and even revealed the previously undetected Wnt3a to be FoxO dependent. This result, combined with the transcriptome analysis and the changed tentacle formation after ALP treatment place insulin and FoxO signaling upstream of the Wnt pathway. To include these findings into the manuscript we added lines 259-266 to the text.

5. The authors propose that the activity of the TGF- β pathway is under the control of Wnt, as proposed by Watanabe et al in 2014 (should be cited for this point). To support this claim, the authors need to provide functional evidences. The results shown in supplemental figure 8 rather suggest that Wnt signaling negatively regulates TGF- β signaling.

Thank you for this very important point. It is true that we propose that the TGF- β pathway is under the control of the Wnt signaling, but we are still unsure about the direction of regulation of individual components. Thus, we do support the claim of Watanabe *et al.*, that Wnt controls TGF- β signaling, but not necessarily that it fulfills an activating function. We were able to show a negative regulation of TGF- β components after an ALP treatment, which highlights the action of Wnt signaling upstream of the TGF- β pathway. We cited Watanabe *et al.* for this reason two times throughout the text and added now a third reference at the mentioned text passage. However, we agreed that evidence for this hierarchy was sparse and performed two additional experiments to provide proof for the acclaimed pathway interaction. First, we could rescue the previously described β -catenin induced multi-headed phenotype of *Hydra*³ and overwrite the constitutively activated Wnt signaling with the TGF- β inhibitor K02288. Second, we were able to rescue an ALP induced Wnt-signal, which led to small sized animals, and even generated large sized polyps using LDN-193189. Both experiments show the rescue of Wnt induced phenotypes by the application of TGF- β receptor inhibitors and demonstrate the hierarchical interaction of the pathways, where Wnt signaling is upstream of TGF- β signaling and both pathways are able to induce size changes in *Hydra*. We added the results of the experiment in a new Figure of the manuscript (Fig. 7, while original Fig. 7 became Fig. 8) and a Supplementary Fig. S14, accompanied with two new paragraphs in the results section (Lines 376-395) and description of the experiment in the methods section (Lines 502-503, 563-564, 572-575).

6. Figure 4b requires better explanation to adequately evaluate the data. Again, it is difficult to understand why the GFP negative control animals of the different KD lines differ so much in size and tentacle number after ALP treatment. The presented results indicate that these controls are not equivalent to each other and also not equivalent to the AEP F2A4 wild-type animals.

Thank you for this comment. Reviewer #1 had a similar question and we added a figure in the supplements (Supplementary Fig. S10) which shows the data set in more depths. We hope that the corresponding explanatory caption and the editions in the main text of the manuscript (lines 294-309) help understanding the results which supports our conclusions. Again, we would like to stress that the size differences of the different transgenic control lines and the wildtype AEP

F2A4 is due to differences in the genetic background of these animals and that different genotypes (lines) show specific maximum sizes. The control lines of INSR-KD C1-1 and FoxO-KD D11a are smaller compared to AEP F2A4 by chance and we added two additional lines (INSR-KD C6-2 and FoxO-KD E11), that show same phenotypes but where the control lines are of similar size as the wild type animals, thus equivalent (Supplementary Fig. S7a, Supplementary Fig. S10a). However, we do not believe that animals with different genetic backgrounds have to be equal in size. This is the reason we chose the corresponding control line to each KD line to compare to and not the AEP F2A4 line.

7. The authors show opposite effects of the two TGF- β receptor inhibitors: a reduction in the number of epithelial cells with K02288 and an increase with LDN-193189. However, the specificity of these drugs in *Hydra* is not tested. Also their respective cellular effects is not tested, as for example a possible increase in apoptosis upon K02288 exposure. In addition, there is no indication on the temperature(s) where these drug treatments were performed. This information is needed given the temperature impact on animal size.

Thank you for this important comment. We performed detailed sequence based *in-silico* analysis to provide good evidence for target specificity and an explanation for the contradictory phenotypes induced by the inhibitors (Supplementary Fig. S12) On the one hand, we show that the ATP binding domain is well enough conserved from *Hydra* to human to serve as specific target for both inhibitors and on the other hand we reveal that differences in the ATP binding pocket of human and *Hydra* TGF- β receptors explain different target specificities of the K02288 and LDN-193189, which gives an explanation for the opposite effects of the inhibitors. We included the results in Supplementary Fig. S12 and in line 329-333 and 595-607 in the manuscript. Further we want to point out that all experiments were performed at 18 °C, unless stated otherwise. However, we added “at 18 °C.” (line 561) in the respective method section which describes the inhibitor experiments. It is true that no cellular effects are tested for either of the inhibitors, but apoptosis would not explain size difference in the K02288 treated animals for several reasons. First, *Hydra* shows only very few cells in an apoptotic state⁴ and if apoptosis is induced they show signs of intoxication like tentacle retraction, which we did not observe during our experiments (or if we did, was a reason to reduce the concentration of the inhibitor). Second, if K02288 would induce apoptosis we would observe a prolongation or at least no difference in the developmental time (Fig 6b) before bud initiation compared to untreated polyps. Third, the mere mechanics of proliferation and apoptosis, which are exponential in nature, would not provide a system which is stable enough to maintain a specific maximum size (a manuscript explaining these effects in detail is in preparation).

Minor points:

- lines 335 and 338: the authors write that the increase in time before budding explains the increase in size of the KD animals, referring there to Figure 6. This is misleading as Figure 6 only contains information about dT and the graphs representing the animal size are in Figure 5.

Thank you, we added a reference to Fig. 5 in the mentioned passage of the text.

- Line 117: Please clarify how the 20 genes were selected.

The 20 genes were representatives of the *Hydra* transcriptome which were previously cloned and sequenced in the lab and therefore verified by experimental evidence.

- In lines 256 and 260 the authors refer to the Supplementary Figure S4 that should represent the fractions of orphan genes in the analysis. This figure does not seem to exist.

Thank you, we are sorry for the inconvenience and added the right figure in the supplements.

- Accession numbers are missing. All raw sequencing data should be made available.

Thank you, it is correct that we forgot to include the SRA accession number. We included the information in the methods section (lines 541-543). The data will be released upon publication but the SRA provides reviewer links to the metadata. Please see below:

ftp://ftp-

trace.ncbi.nlm.nih.gov/sra/review/SRP133389_20190125_113520_37d5c0b6b354bc3c790d2696b42756c9

• The references for Supplementary Figure S2 and S3 are inverted in the text.

Thank you, we corrected the references in the text.

References:

1. Wittlieb, J., Khalturin, K., Lohmann, J. U., Anton-Erxleben, F. & Bosch, T. C. G. Transgenic Hydra allow in vivo tracking of individual stem cells during morphogenesis. *Proc Natl Acad Sci U S A* **103**, 6208–6211 (2006).
2. Franzenburg, S. *et al.* Distinct antimicrobial peptide expression determines host species-specific bacterial associations. *Proc Natl Acad Sci U S A* **110**, E3730-8 (2013).
3. Gee, L. *et al.* β -catenin plays a central role in setting up the head organizer in hydra. *Dev Biol* **340**, 116–124 (2010).
4. Bosch, T. C. & David, C. N. Growth regulation in Hydra: relationship between epithelial cell cycle length and growth rate. *Dev Biol* **104**, 161–71 (1984).

Reviewer #3 (Comments to the author):

Methodologically sound in general but can the authors clarify how they controlled for multiple statistical

comparisons that may lead to a Type I error (i.e. false positive findings) ?

Thank you for this comment and we are happy to elaborate our statistical methods. We stated in the methods section that we used either the two-tailed Student's-t-test or the Mann-Whitney-U-test, where it was applicable, in order to test for statistical significance between the treatments. We agree that controlling for multiple comparisons is an important and often neglected part of biostatistics. Especially when dealing with next generation sequencing data, p-value correction is very important in order to distinguish significant differences from false positive findings and artifacts. Therefore, wherever multiple testing occurred in our data, we adjusted the p-values using the Benjamini-Hochberg correction¹.

The discussion should try to tie in the human work in this area to make the findings more relevant across species. While the authors briefly mention genetic influence over human body size there is a body of work on FOXO3, body size and longevity in humans that support the manuscript's current findings and could strengthen the conclusions. This deserves to be mentioned in another paragraph or two in the discussion. Some relevant papers appear below.

- Q He, BJ Morris, JS Grove, H Petrovitch, W Ross, KH Masaki, B Rodriguez, ...Shorter Men live Longer...PLoS One 2014; 9 (5), e94385.

-Bartke A. Somatic Growth, Aging and Longevity. NPJ Aging Mech Dis. 2017 Sep 29;3:14. doi:10.1038/s41514-017-0014-y. eCollection 2017

- Davy P *et al.* FOXO3 and Exceptional Longevity: Insights From Hydra to Humans. *Current topics in developmental biology* 2018:127, 193-212.

- Samaras TT. How height is related to our health and longevity. *Nutr Health*. 2012 Oct;21(4):247-61. doi: 10.1177/0260106013510996. Review.

- Nebel and Bosche. Evolution of human Longevity. *Lessons from Hydra*. Aging (Albany NY). 2012 Nov;4(11):730-1.

Thank you for this suggestion and we agree that FoxO related work on humans is a bit underrepresented in the discussion. Therefore, we added a small text passage referencing the latest literature in the FoxO field (lines 430-433). However, we feel the scope of this manuscript is not to give detailed insights into FoxO research which was discussed in numerous excellent reviews in the last few years.

Minor Issues

- Line 319: the word “neither” should be “either.”

Thank you, we corrected the typo.

References:

1. Benjamini, Y. & Hochberg, Y. Controlling the false discovery rate: a practical and powerful approach to multiple testing. *J R Stat Soc* 57, 289–300 (1995).

At this point, we wish to thank the reviewers for the constructive and thoughtful comments. They improved the manuscript greatly.

Reviewers' Comments:

Reviewer #1:

Remarks to the Author:

The authors have made improvements to the manuscript and have addressed many concerns raised by the reviewers. However, I do not support the publication at this stage, because there remain the issues raised by the reviewer that still were not sufficiently addressed.

1. Fig. 5: To establish that insulin signaling controls hydra's size through TGF-b, the authors were suggested to examine the effects of perturbation of insulin signaling on the animal size in the presence of TGF-b inhibitors. I see that only negative controls (Control KD + No drug) were added in the revised Fig. 5 c,d,e. These panels should also contain "Control KD + Drug", enabling total four lanes to be compared. To support the authors' claims, in principle, the size of "Control KD + Drug" and that of "INSR KD + Drug" should be similar while the size of "Control KD + No drug" is smaller than "INSR KD + No drug". Likewise, the size of "Control KD + Drug" and that of "Foxo KD + Drug" should be similar while the size of "Control KD + No drug" is bigger than "Foxo KD + No drug". Similar changes by the KD treatment in the presence and in the absence of the drug indicate that the KD treatment affects body size regardless of the drug.

2. The authors claim that the potential off-targets of insR-hairpin, selected by BLAST search, did not show decreased expression, trying to exclude the possibility of off-target effects. Unfortunately, it is virtually impossible for the BLAST to cover the whole transcriptome of the animal under every possible environmental and physiological conditions, and thus the authors' experiments cannot exclude the possibility that there might be off-target genes in the group of downregulated genes by the knockdown experiment. Even more problematic, the authors confess that they cannot monitor the change of the intensity of insulin signaling in the animal treated by the knockdown. In this situation, it is hard to be assured that the description and analysis of the data presented are really from decreased insulin signaling. The author should consider additional ways of decreasing insulin signaling to confirm the effects of insR-hairpin.

3. I'm still confused about the inhibitors of TGF-b used in the study. The two inhibitor showed opposite effects in Fig5. Do the authors claim that inhibition of TGF-b can lead these contradictory results? Did the authors check that these two drugs really inhibited the TGF-b signaling in the experimental conditions? Is it more reasonable to think that one of them acts as an inhibitor while the other as an activator of TGF-b signaling? The authors should clarify the biological effects of these two drugs.

Reviewer #2:

Remarks to the Author:

The authors have satisfied our requirements

Our responses and changes to the manuscript are given in **bold**.

Referee #1 (Remarks to the Author):

1. Fig. 5: To establish that insulin signaling controls hydra's size through TGF- β , the authors were suggested to examine the effects of perturbation of insulin signaling on the animal size in the presence of TGF- β inhibitors. I see that only negative controls (Control KD + No drug) were added in the revised Fig. 5 c,d,e. These panels should also contain "Control KD + Drug", enabling total four lanes to be compared. To support the authors' claims, in principle, the size of "Control KD + Drug" and that of "INSR KD + Drug" should be similar while the size of "Control KD + No drug" is smaller than "INSR KD + No drug". Likewise, the size of "Control KD + Drug" and that of "Foxo KD + Drug" should be similar while the size of "Control KD + No drug" is bigger than "Foxo KD + No drug". Similar changes by the KD treatment in the presence and in the absence of the drug indicate that the KD treatment affects body size regardless of the drug.

Thank you for the comment. Obviously we have not made clear enough the nature of our knock-down (KD) control lines. In order to remove any doubt, we explain our used KD control lines more thoroughly: Each of our *Hydra* KDs, being INSR-KD or FoxO-KD, comes with its own KD control line. After microinjection of DNA plasmids into early embryos, we obtained transgenic lines that observe a mosaic pattern after hatching. By continuous selection for transgenic cells in the ectodermal and/or endodermal epithelium during asexual reproduction, we were able to generate completely transgenic animals for these tissues. However, at the same time, we also selected for non-transgenic animals and thereby created KD line-specific controls. To provide this information for the reader, we added explanations in the manuscript at lines 143-148 and 459-465.

Therefore, our wild type line in Fig. 5a and b has a different genetic background than the INSR-KD control in Fig. 5c or the FoxO-KD control line in Fig. 5d and e. As discussed in the first revision, all these lines have different maximum sizes. However, all lines respond to the used inhibitors similarly as our wild type line as shown in Fig. 5a and b. In order to reduce the number of factors to be compared and make the figure more comprehensible, we decided to not include these data in all of the plots shown. Therefore, we decided to include an alternative version of Fig. 5c, labeled as Supplementary Figure 12, into the revised manuscript. The plot includes the data for the treatment of the INSR-KD control line with K02288. As you pointed out, it is obvious that treatment of FoxO-KD with LDN-193189 is not able to rescue the phenotype of the knockdown completely, which may suggest that treatment with LDN-193189 is blocking the TGF- β signaling incompletely, as we observed toxic effects of the LDN-193189 at lower concentrations than in the wild type line. Further, we do not imply to know the detailed regulation of TGF- β on the size of *Hydra* polyps as there are several different ligands (including TGF- β s, BMPs, GDF, Activin, Inhibin, and Nodal) which might bind to about 9 different type I TGF- β receptors and at least 4 different type II TGF- β receptors in *Hydra* with different affinities and signaling outcomes (like inhibition or activation). The situation is further complicated by co-receptors (TGF- β receptor 3) and modulating proteins (e. g. DAN, NBL, Cerberus). Thus, it may be, that other parts of the TGF- β pathway, which are not inhibited by LDN-193189, eventually compensate for the inhibited parts. However, the incomplete rescue of the corresponding knockdown prompted us to skip further experiments with the FoxO-KD control line, as we have already shown that the inhibitor works in a wild type context.

2. The authors claim that the potential off-targets of insR-hairpin, selected by BLAST search, did not show decreased expression, trying to exclude the possibility of off-target effects. Unfortunately, it is virtually impossible for the BLAST to cover the whole transcriptome of the animal under every possible environmental and physiological conditions, and thus the authors' experiments cannot exclude the possibility that there might be off-target genes in the group of downregulated genes by the knockdown experiment. Even more problematic, the authors confess that they cannot monitor the change of the intensity of insulin signaling in the animal treated by the knockdown. In this situation, it is hard to be assured that the description and analysis of the data presented are really from decreased insulin signaling. The author should consider additional ways of decreasing insulin signaling to confirm the effects of insR-hairpin.

We agree that the newly assembled transcriptome against which, we performed the BLAST search, may not represent every transcript ever produced in any possible environmental or physiological condition. However, that is why we used the INSR-KD control line to account for conditions we cannot control throughout our experiments as a reference point. Any change we observed in INSR-KD animals, which is induced by the INSR-KD construct (on- or off-target) should be visible if comparing to the INSR-KD control line under the given conditions. Even in the improbable case, where a transcript was not assembled (i. e. was not expressed in our experimental conditions) but would have been present in the INSR-KD condition (which would indicate an up-regulation of this transcript upon the expression of the hairpin construct), would be hardly proof for an off-target effect, as we would expect a down-regulation of the off-target transcript in addition to the INSR transcript by the hairpin construct. Furthermore, we used very liberal search criteria to include all possible off-targets, as improbable as they may be, and still are unable to identify a transcript, which could explain the observed phenotype, except the knockdown of the INSR.

We agree that we have no means showing the intensity of the insulin signaling in *Hydra*, except the reduction of transcript levels of the INSR and FoxO and the, in the end, biological important and consistent phenotype of altered size by cell number changes. It is impossible for us, to perform all experiments needed to proof the interactions, functionality and regulation of every component of the insulin pathway from ligand to effector in *Hydra* which was described in other model organisms and is, as we believe, far beyond the scope of a single (this) paper. But this is what would be requested, if on the one hand, it is doubted that the INSR is signaling input for FoxO regulation (see previous revision), and on the other hand it is demanded for additional ways of decreasing the insulin signaling.

We provided two targets in the insulin pathway with a size phenotype, provided replicates for the KD lines and assembled a new transcriptome for this study in order to identify all changes in gene expression induced by our treatments. All our findings are very consistent within *Hydra* and we have no reason to believe that off-target effects played a role in any of our data sets.

3. I'm still confused about the inhibitors of TGF-b used in the study. The two inhibitor showed opposite effects in Fig5. Do the authors claim that inhibition of TGF-b can lead these contradictory results? Did the authors check that these two drugs really inhibited the TGF-b signaling in the experimental conditions? Is it more reasonable to think that one of them acts as an inhibitor while the other as an activator of TGF-b signaling? The authors should clarify the biological effects of these two drugs.

We explain the different effects of LDN-193189 and K02288 on the size phenotype by the different inhibitor affinities (see Supplementary Figure 11). We argue that the two inhibitors have

distinct targets in the TGF- β -receptor family of *Hydra*, that is: LDN-193189 inhibits ACVR1, while K02288 inhibits TGFR1 most efficiently. We explained this shortly in the manuscript in lines 293-299 and more thoroughly in the caption of Supplementary Fig. 11. This is different to the human or mouse situation and explains how inhibitors with virtually exact same receptor affinities (measured in human or mouse) have different effects on the body size in *Hydra*. As explained above, the TGF- β pathway is diverse in ligands (TGF- β s, BMPs, GDF, Activin, Inhibin, Nodal etc.) and receptors (several type I, and II) alike, further complicated by ligand-receptor binding modulators, co-receptors and the possibility of each ligand to bind to several receptors with different affinities and *vice versa*. We thus tested several TGF- β receptor inhibitors for effects on size in *Hydra*, including the reported LDN-193189 and K02288 as well as Dorsomorphin, A83-01, iCRT14, LDN-212854, NSC668036 and Niclosamide. Most of the tested substances were toxic or showed no effect in subtoxic concentrations, but we found the two reported TGF- β receptor inhibitors and their potency to change the size of the polyps. We therefore focused here on the TGF- β pathway as a candidate pathway for size regulation in *Hydra*. Efforts to disentangle the effects of each ligand-receptor interaction and their specific role in bud initiation and thus size determination in *Hydra* goes far beyond the scope of this manuscript.